# A Highly Sensitive and Selective ppb-Level Acetone Sensor Based on a Pt-Doped 3D Porous SnO_2_ Hierarchical Structure

**DOI:** 10.3390/s20041150

**Published:** 2020-02-19

**Authors:** Wenjing Quan, Xuefeng Hu, Xinjie Min, Junwen Qiu, Renbing Tian, Peng Ji, Weiwei Qin, Haixin Wang, Ting Pan, Suishi Cheng, Xiaoqiang Chen, Wei Zhang, Xiaoru Wang, Hua Zheng

**Affiliations:** 1State Key Laboratory of Materials-Oriented Chemical Engineering, College of Chemical Engineering, Nanjing Tech University, No. 5 Xin Mofan Road, Nanjing 210009, China; 714403200@njtech.edu.cn (W.Q.); mxj307414610@njtech.edu.cn (X.M.);; 2School of Instrument Science and Opto-Electronics Engineering, Research Center for Sensor Science and Technology and Special Display and Imaging Technology Innovation Center of Anhui Province, Hefei University of Technology, No. 193 Tunxi Road, Hefei 230009, China; xuefeng.hu@hfut.edu.cn; 3School of Electrical Engineering & Intelligentization, Dongguan University of Technology, No.1 Daxue Rd, Dongguan 523808, China; 4Anhui 6D Sensing Technology Co., Ltd., Yingtian Industrial Park, Fuyang City 236000, China

**Keywords:** ppb-level acetone sensor, hierarchical flower-like SnO_2_ structure, precious Pt-doped porous structure, diabetes diagnosis

## Abstract

In view of the low sensitivity, high operating temperature and poor selectivity of acetone measurements, in this paper much effort has been paid to improve the performance of acetone sensors from three aspects: increasing the surface area of the material, improving the surface activity and enhancing gas diffusion. A hierarchical flower-like Pt-doped (1 wt %) 3D porous SnO_2_ (3DPS) material was synthesized by a one-step hydrothermal method. The micropores of the material were constructed by subsequent annealing. The results of the experiments show that the 3DPS-based sensor's response is strongly dependent on temperature, exhibiting a mountain-like response curve. The maximum sensor sensitivity (R_a_/R_g_) was found to be as high as 505.7 at a heating temperature of 153 °C and with an exposure to 100 ppm acetone. Additionally, at 153 °C, the sensor still had a response of 2.1 when exposed to 50 ppb acetone gas. The 3DPS-based sensor also has an excellent selectivity for acetone detection. The high sensitivity can be explained by the increase in the specific surface area brought about by the hierarchical flower-like structure, the enhanced surface activity of the noble metal nanoparticles, and the rapid diffusion of free-gas and adsorbed gas molecules caused by the multiple channels of the microporous structure.

## 1. Introduction

Acetone is a well-known solvent and has been widely used in various industries, laboratories and home applications [1,2]. Acetone is a volatile, flammable and deleterious compound and is harmful to human health; even prolonged exposures to low concentrations of acetone (<1 ppm) can still cause headaches, nausea, muscle weakness, loss of coordinated speech, narcosis and harm to the nervous system [3,4]. More interestingly, the latest research shows that early diabetic patients may exhale more than the standard amount of acetone gas, and thus, the early detection and screening of high-risk diabetic patients can be fulfilled only by detecting the amount of acetone in the exhaled gas. The advantages of this innovative diagnostic method are that it is noninvasive, easy to use and inexpensive [5]. Although the acetone concentration necessary to diagnose diabetes, according to the verified standard, is currently recognized to be greater than 1.8 ppm [6], medical evidence shows that for very early diabetic patients, the amount of acetone in the exhaled gas could be at the ppb level. Currently popular and optional methods used to measure low concentrations of acetone are gas chromatography and mass spectrometry (GC-MS). These two methods demonstrate a high measurement sensitivity and stability, but these methods are not portable and are expensive, complicated to operate and time-consuming. Another problem encountered in the process of achieving a diabetes diagnosis by measuring acetone in the breathing gas is determining how to avoid interferences from other gases. A diabetic patient may have other diseases at the same time, which correspond to other marker gases in the breath. Thus, the portable sensors used to accurately measure trace levels of acetone in the breathing gas must have a high sensitivity and selectivity.

Acetone gas sensors based on metal oxides semiconductors (MOSs) have attracted the most attention due to their simplicity of operation, low cost, and fast response/recovery speed, as well as their high sensitivity. The gas-sensing effect is usually a surface phenomenon, i.e., the changes in the concentration of conduction electrons in metal oxide semiconductors result from gas surface chemisorption, reduction/oxidation ("redox"), and/or catalysis processes [7,8,9]. Any means of influencing the surface state can be used to modify the metal oxide or semiconductor surface. MOSs-based sensors are generally so dense that gas diffuses into the underlayer, making absorption and reactions impossible. Thus, only the surface of the sensing film participates in the reaction and contributes to the sensor response. An intuitive method used to increase the sensor sensitivity is to increase the ratio of the surface area to the volume of the material. The sensitivity and selectivity of the sensor also depend greatly on the surface properties of the sensing material. Any attempt to improve these characteristics, such as doping with precious metal particles, can effectively improve the performance of the sensor.

Periodically assembled, hierarchical metallic oxides have large surface areas that are advantageous for chemical reactions, allow for chemical species (ions or gases) to effectively diffuse into the interfaces/surfaces, and perform well as gas sensors, and thus, these materials have spurred great attention [10]. A "hierarchical structure" means that the higher dimension of a micro- or nanostructure are composed of many, low dimensional, nanosized building blocks. The van der Waals attractions between hierarchical structures are relatively weak [11], because the hierarchical structures are generally larger than the individual nanostructures. Thus, hierarchically assembled microspheres are more flowable than nanowires and nanosheets.

Many studies have focused on the synthesis of hierarchical structures and gas sensor applications. Y. Liu et al. [12] reported the hydrothermal synthesis of hierarchical SnO_2_ nanostructures, which exhibited an excellent sensitivity to ethanol gas at a temperature of 300 °C. Sun et al. [13] also fabricated SnO_2_ hierarchical architectures by a hydrothermal synthesis method, and these SnO_2_ structures demonstrated a superior high response to ethanol at the optimal operating temperature of 275 °C. Wei et al. [14] successfully synthesized flower-like SnO_2_ hierarchical nanostructures using SnSO_4_ as the tin source and water as the solvent, and these nanostructures exhibited good n-butanol gas sensing properties at 160 °C. Sensing materials with porous structures, which create more channels for gas diffusion and reactions, have also been synthesized [15]. Noble metals such as Co [16], Au [17,18], Pt [19,20] and Ag [21] have been used to decorate the surface of SMOs to enhance gas adsorption and lower the reaction activation energy.

Specifically, in the field of acetone sensor applications, Liu et al. [22] synthesized peony-like hierarchical Sb-doped In_2_O_3_ flowers with different Sb contents and obtained a response (R_a_/R_g_, where R_a_ is the initial resistance of the sensor and R_g_ is the resistance under an exposure the gas) of 64.3 at an acetone concentration of 50 ppm and a heating temperature of 240 °C. Kim et al. [23] developed a vertically ordered hematite nanotube array to improve the gas sensing response to 84 at 50 ppm acetone but at a higher working temperature of 350 °C. Au@ZnO yolk-shell nanospheres were developed by Li et al. [24] and exhibited a sensing response of 37 at 100 ppm acetone and at 325 °C. While great efforts have been directed toward developing improved acetone gas sensors, the performance of these sensors is still far from what is needed to detect trace acetone for the early diagnosis of diabetes [25]. First, the sensitivity of these sensors is not adequate for detecting low concentrations, especially at the ppb level, and further increasing the sensor’s sensitivity still remains a challenge. Second, the response time of the sensor is up to several hundred seconds, which is not suitable for rapid and in situ measurements. Furthermore, the elimination of interferences from other gases in the breathing gas presents a significant challenge, as it is difficult for the sensor to isolate the acetone target. Finally, theoretical issues related to the gas adsorption process, sensor performance, catalytic mechanism of noble metal atoms, role of the pore structure and optimal operating temperature require further study and understanding.

In this article, to improve the properties and clarify the mechanism driving the sensing capability of sensitive gas sensing materials, three SnO_2_ sensor materials with similar microstructures, 3D hierarchical SnO_2_ microflower structures, Pt-doped (1 wt %) 3D SnO_2_ (3DS), and Pt-doped (1 wt %) 3D porous SnO_2_ (3DPS), were all synthesized by a one-step hydrothermal method. Basically, SnO_2_ was selected for this study due to its high sensitivity to acetone gas and ability to operate at low temperatures [26,27]. The sensitivity of the acetone sensor is anticipated to improve by combining the synthesized hierarchical structures to increase the specific surface area and modifying the surface of the structures with precious metals to enhance the reaction of the adsorbed gas. A post-annealing treatment is used to create micropores in the SnO_2_ hierarchical structure, which desirably increase the gas diffusion rate and shorten the response time. The enhancement of the sensor selectivity can be achieved by selecting suitable metal oxide materials and modifying with precious metals. The full characterization of the pure SnO_2_, Pt-doped 3DS and Pt-doped 3DPS microstructures and their gas responses have been performed by various electrical, structural and surface analytical tools. Finally, the mechanism leading to the high sensitivity, high selectivity and optimum operating temperature of these structures is studied and explained.

## 2. Materials and Methods

### 2.1. Materials

Stannous chloride (SnCl_2_·2H_2_O, 97 wt %), sodium hydroxide (NaOH, 97 wt %), salicylic acid (C_6_H_4_OHCOOH, 97 wt %), and anhydrous ethanol (C_2_H_5_OH) were of analytical grade. H_2_PtCl_6_·6H_2_O was purchased from Sinopharm Chemical Reagent Co., Ltd. (Shanghai, China) and was used as received. All the aqueous solutions were prepared using 18.2 MΩ ultrapure water.

### 2.2. Synthesis of the Pure SnO_2_ Microflowers

A simple one-pot low-temperature hydrothermal strategy was used to synthesize the SnO_2_ microflowers. To prepare the microflowers, 1.78 mmol of SnCl_2_·2H_2_O was dissolved in 10 mL deionized water, and 9 mmol NaOH was added into the mixed liquid (named solution A). Then, 3 mmol salicylic acid was dissolved in a 20 mL mixture of deionized water and anhydrous ethanol (volume ratio was 1:1) (named solution B). Solution B was slowly added to solution A at one drop per five seconds under constant stirring. When half of the salicylic acid solution was added, the addition of the remaining solution B was paused so that the mixture could be stirred for 30 min. Then, the remaining solution B was added using the abovementioned conditions. When the addition was completed, the mixture was stirred constantly by magnetic stirring for 30 min. Then, the turbid liquid was transferred to a 50 mL Teflon autoclave and heated at 150 °C for 16 h. After the autoclave was cooled naturally to room temperature, the brownish yellow precipitate was collected by centrifugation and washed six times with deionized water and anhydrous ethanol, and then, the precipitate was dried in air at 60 °C for 4 h prior to further application.

### 2.3. Synthesis of the Pt-Doped 3DS and Pt-Doped 3DPS Microflowers

To prepare the microflowers, 1.78 mmol SnCl_2_·2H_2_O was dissolved in 10 mL deionized water, and then, 9 mmol NaOH and H_2_PtCl_6_·6H_2_O (Pt/Sn = 1 wt %) were added into the mixed liquid (named solution A). Then, 3 mmol salicylic acid was dissolved in a 20 mL mixture of deionized water and anhydrous ethanol (volume ratio was 1:1) (named solution B). Under the same abovementioned experimental conditions, the final product was dried at 60 °C for 4 h to obtain Pt-doped 3DS. To examine the effect of pores on the sensor properties, pristine Pt-doped 3DS was calcined in air at 700 °C for 2 h with a ramping rate of 5 °C·min^−1^ to obtain the final Pt-doped 3DPS microflowers.

### 2.4. Characterization of the Sensing Materials

X-ray diffraction (XRD, D8-Advance, Bruker, Germany) was used to characterize the structure of the sample with focused monochromatized Cu-Kα radiation with a wavelength of 0.15418 nm. Data were collected in the 2θ range of 20°–80° at a rate of 20°·min^−1^. The average d-spacing value was calculated from the X-ray data using Bragg’s law. Energy dispersive X-ray spectroscopy (EDX, Quanta 200, FEI, U.S.A) was utilized to confirm the composition of the materials. The surface chemical composition and valence state of the sensing materials were determined by X-ray photoelectron spectroscopy (XPS, Thermo ESCALAB 250Xi) with a monochromatic Al-Kα X-ray source (1486.6 eV). The morphology and size of the particles were determined by field emission scanning electron microscopy (FE-SEM, Quanta 200, FEI, U.S.A). The nanoscale images of the samples were obtained by transmission electron microscopy (TEM, Tecnai G2 F30 S-TWIN, FEI, American) using a microscope operated at 200 kV. The textural properties of the as-obtained powders, including the Brunauer-Emmett-Teller (BET) surface area and pore properties, were analyzed by a low temperature N_2_ adsorption/desorption method using a Micromeritics ASAP 2390 volumetric adsorption analyzer that was predegassed at 120 °C for 16 h.

### 2.5. Fabrication and Characterization of the Gas Sensor

The SnO_2_ powder materials were dispersed in ethanol to form a colloid, and the colloid was then coated on an alumina chip (area = 1.0 × 1.5 mm, thickness = 0.2 mm). The sensor chip, as-shown in Figure 1, was designed with 4 electrodes: two electrodes for heating and two electrodes for detection. The heating and probing electrode was electrically separated by a thin insulating SiO_2_ film. The sensor component was then wire bonded with a standard Transistor Out-line (TO) package. The heating temperature was adjusted from 109 to 240 °C. The gas sensing properties were measured by an Integrated Gas Sensing Test System (LW-GS-002, Anhui Six-dimension Sensor Technology, Ltd.) with two mass flow meters (MFs) to control the acetone gas concentration (50 ppb−2 ppm), and high purity air was used as the diluent and carrier gas. Different concentrations of acetone gas (from 10 ppm to 1000 ppm) were injected into the test chamber (25 mL in volume) through a rubber plug by a microinjector to form an acetone vapor. The saturated target gas was mixed with air (where the relative humidity of 35% was measured at 25 °C) by natural diffusion. The resistance of the materials was measured and collected in real-time with a time step size of one second. The gas sensing response is defined as the ratio (S = R_a_/R_g_) of the resistance of the sensor in air (R_a_) to that under an exposure to the target gases (R_g_). The response and recovery times are defined as the time taken by the sensor to achieve 90% of the total resistance change in either the atmospheric air or target gases. Note that the changes in the sensor response times are recorded for the gas concentrations, ranging from zero to the set concentration. The changing response times actually depends not only on the performance of the sensor but also on the test equipment. Thus, all the response and recovery times obtained in the real-time measurements are only meant as references.

## 3. Results

### 3.1. Structure, Morphology and Composition of the Sensing Materials

The structure of the synthesized pure SnO_2_, Pt-doped 3DS and Pt-doped 3DPS samples, as shown in Figure 2, were characterized by XRD. The results of the diffraction peaks in the 2θ range from 20° to 80° verified that all three of the prepared samples are standard tetragonal rutile crystal phases of SnO_2_ (JCPDS file No. 41-1445, a, b = 0.474 nm and c = 0.319 nm), indicating that Pt doping and high-temperature post-annealing does not affect the hierarchical structure. For the calcined Pt-doped 3DPS sample, the sharpness of the peaks suggested an improved crystallinity. No obvious diffraction peaks corresponding to Pt were observed, since the Pt particles were too small or too highly dispersed to be detected.

The morphologies of the three hydrothermally synthesized samples, as shown in Figure 3, were observed by SEM, where the images on the left and right of Figure 3 are the low and high magnification images, respectively. The low magnification SEM image in Figure 3a,c,e shows the general morphology of all three products, in which a 3D hierarchical nanostructure can be observed and is composed of numerous flower-like microstructures with diameters of 2–4 μm. These flowers are self-assembled by many uniform 2D nanosheets that assemble at the flower centers and extend toward different directions. The high magnification SEM image in Figure 3b,d,f clearly demonstrates that the nanosheets have a thickness of approximately 10–40 nm. From Figure 3f, after calcining with the Pt dopant, it is obvious that the microstructure contains many voids and interspaces among the petals to form a unique low density and porous 3D structure, and these voids and interspaces not only provide an excellent channel for the transport of detected gases but also serve as reaction centers for the target gas. The thermal decomposition of the organic impurity (salicylic acid) and the release of gas into the confined space during the sintered process have resulted in the formation of the porous structure. As evidenced by Figure 3d, a smooth surface was observed for the sample prepared without calcining. Thus, the calcination process is a key factor for the formation of pores. Figure 3g also shows the corresponding EDX spectra of Pt-doped 3DPS. Only peaks corresponding to Pt, Sn and O elements were observed, suggesting that Pt was successfully doped in the sample and high purity samples were obtained. The amount of Pt doping is estimated to be 1 wt % from the weight ratios of Pt to Sn.

To analyze the fine microstructure of the Pt-doped 3DPS sample in nanoscale, TEM and high-resolution TEM (HRTEM) characterizations were utilized. The low-magnification TEM images, as shown in Figure 4a,b exhibited that the size of Pt-doped 3DPS was several micrometers, and the morphology of the sample was similar to that determined by the SEM observations. Pores with irregular shapes can also be seen in the nanosheet, as shown in Figure 4b, which is potentially beneficial for gas diffusion and the transport of gas molecules in the sensors. As seen from the magenta area in Figure 4b, PtO_2_ was uniformly dispersed on 3DPS. To further observe the SnO_2_ 3D microstructure and distribution of Pt and PtO_2_ nanoparticles in the Pt-doped 3DPS microstructure, HRTEM was used. The HRTEM images, as shown in Figure 4c,d revealed that SnO_2_ had a single crystal structure; the d-spacings were determined to be 0.335 nm and 0.264 nm, which agreed well with the d-spacing of the (110) and (101) planes of the tetragonal cassiterite structure of SnO_2_. The distance between the adjacent lattice planes is estimated to be 0.223 nm, which corresponds to the distinct PtO_2_ (111) lattice fringes shown in Figure 4c,d and demonstrates the coexistence of Pt. According to the size distribution diagram (Appendix A), the average size of nanoparticles is estimated to be about 8.7 nm. Post-annealing promotes the formation of metallic clusters, which improve the homogeneity of their distribution across the layer thickness and stabilize the solid-phase of the Pt-doped 3DPS material. However, increasing the annealing temperature further may result in the agglomeration of Pt and PtO, reducing the activity of the material and ultimately reducing the sensitivity of the sensor.

To investigate the chemical state of the elements and chemical composition of the materials, XPS analysis was performed, and the results are shown in Figure 5. Figure 5a shows the high-resolution XPS analysis for the full range of the spectrum. The peaks corresponding to Sn, Pt and O, as well as C, can be clearly observed in the survey spectrum. The C 1s binding energy of 284.7 eV was used as the reference for calibration. The weak C peak is most likely due to surface contamination that occurred during sample handling and storage. One unidentified peak at 1062 eV may be attributed to the Auger transitions. The binding energies correspond to different energy levels of O, Sn and C and are summarized in Figure 5a. There are distinct peaks corresponding to Sn (4d), Sn (3d), Sn (3p), O (1s) and Pt (4f). Figure 5b presents the XPS spectrum of Sn 3d. Two strong peaks centered at 486.9 and 495.0 eV can be identified as the bonding energies of Sn 3d_5/2_ and Sn 3d_3/2_, respectively, which are attributed to the spin-orbit coupling of the 3d state with the 8.1 eV spin-orbit separation. The result is well-match with the standard data of SnO_2_, proving that Sn^4+^ is present in SnO_2_ [21]. Note that the intensity of the Pt 4f peaks in Figure 5a are almost too low to observe in the full range of the spectrum because of the low concentration of Pt. The Pt 4f (Figure 5c) signals display two pairs of doublets, indicating the presence of two oxidation states. The most intense doublet with binding energy of 73.3 eV (Pt 4f_7/2_) and 76.6 eV (Pt 4f_5/2_) is attributed to Pt^2+^ as in PtO. The second doublet found at 74.1 and 77.6 eV appears to be Pt^4+^, possibly as PtO_2_ [28,29,30]. In contrast, metallic Pt peaks (70-71 eV) were not detected in the crystal structure of Pt-doped 3DPS in Figure 5c [31]. Three peaks at 530.7, 531.4 and 531.7 eV were observed the asymmetric O 1s spectrum, as exhibited in Figure 5d, and can be ascribed to the oxygen species in Pt-doped 3DPS. The lowest binding energy of 530.7 eV corresponds to the SnO_2_ lattice oxygen, whereas the medium binding energy of 531.4 eV is due to the O^2−^ ions in the oxygen vacancy regions. Additionally, the high binding energy located at 531.7 eV could be attributed to the presence of oxygen ions chemisorbed on the SnO_2_ surface. All the above-mentioned qualitative XPS results confirmed that the Pt dopants were incorporated into the Pt-doped 3DPS material. XPS shows a Pt concentration of 0.21%.

To further obtain information about the specific surface area, pore size and pore distribution of the Pt-doped 3DPS microstructure, the nitrogen adsorption and desorption measurements were performed at 77 K. The results of the N_2_ adsorption-desorption isotherm and the corresponding pore-size distribution (PSD) curves of the Pt-doped 3DPS microstructure are shown in Figure 6. The specific surface areas were measured by the Brunauer–Emmett–Teller (BET) method, and the PSDs were calculated from the adsorption branches using the Barrett-Joyner-Halenda (BJH) formula. According to the IUPAC classification, the isotherm obtained for the Pt-doped 3DPS sample can be approximately classified as a type IV isotherm with a type H3 hysteresis loop, indicating the existence of slit-like pores in the materials that formed by the aggregation of sheet-like particles [32]. The presence of highly mesoporous (pores 2–50 nm diameter) structures is suggested by the prominent hysteresis loops obtained at relative pressures from 0.45 to 1.0. Using the BJH method and the adsorption branch of the nitrogen isotherm, the calculated PSD indicated that the material contained pores with average sizes of 10 nm, which agreed with the observations obtained from the TEM and SEM images. The BET surface area of the products was calculated to be 156.72 m^2^ g^−1^. As shown in Appendix A, the pure SnO_2_ and Pt-doped 3DS samples possess specific surface areas of 78.29 and 129.30 m^2^ g^−1^, respectively. The insets of graphs show the pore size of the two samples. No peak was observed in the pore size distribution curves of pure SnO_2_ and Pt-doped 3DS samples. The excellent properties, including the high specific surface area and appropriate pore size distribution, makes the Pt-doped 3DPS material a promising candidate for improving the performance of gas sensors.

### 3.2. The Characterization of Gas Sensor Performance

It is well-known that the response of a semiconductor gas sensor is considerably affected by the working temperature due to significant influence of temperature on all aspects of gas sensing processes, including the gas adsorption/desorption, diffusion and reactions processes. The optimal working temperature of a gas sensor is defined as the temperature corresponding to the maximum response. To explore the influence of temperature, the gas sensing responses of the abovementioned three sensors were characterized at operating temperatures in the range of 109 to 240 °C with 100 ppm acetone. Lower temperatures were not tested in this experiment due to limited electrical power. The electrical power needed to heat the sensor to 109 °C is approximately 45 mW, which satisfied the power requirement for almost all portable devices. The gas sensing performances of the three samples to detect 100 ppm acetone, as shown in Figure 7, exhibit a similar trend. The responses of all the sensors first increased with operating temperature, achieving a maximum value at 153 °C, and then gradually decreased by further increasing the heating temperature. Obviously, 153 °C is the optimum working temperature from the tested temperature values for all three samples, and the highest gas response to 100 ppm acetone is estimated to be 70.9, 143.1 and 505.7 at 153 °C for pure SnO_2_, Pt-doped 3DS and Pt-doped 3DPS respectively. The Pt-doped 3DPS sensor exhibits the highest response to 100 ppm acetone for temperatures less than 153 °C, and this response is over seven times higher than the response of the pure SnO_2_ sensor. The mechanism explaining the optimal temperature for the maximum sensing response is complex [33]. Mainly, this mechanism may be related to the competitive effect of the temperature on the absorption, desorption, reaction and diffusion processes. These details will be discussed later.

To explore the measured gas concentration limit of the gas sensor at the optimum temperature, the sensing responses of the sensors were measured for different acetone concentrations from 1 ppm to 1000 ppm for the three samples: pure SnO_2_, Pt-doped 3DS and Pt-doped 3DPS. Figure 8a,b clearly shows that the gas responses increased monotonically with an increase in the concentration of acetone gas. Because acetone is a reducing gas, the sensor is an n-type semiconductor material, and the resistance of the sensor is reduced when the gas is introduced and returns to the initial value after the introduction of acetone is stopped or fresh air is introduced. The largest response to a 100 ppm acetone concentration was exhibited by the Pt-doped 3DPS material and was approximately seven times higher than the response of the pure SnO_2_ sensor at the same acetone concentration and at an operating temperature of 153 °C. Since measuring low acetone concentrations is a requirement for the early diagnosis of diabetes, Figure 8c shows the measurement of acetone concentrations ranging from 50 ppb to 2 ppm for the Pt-doped 3DPS-based sensor. The minimum concentration was established as 50 ppb because of the limitations of our measuring equipment. A response of 2.1 was found at an acetone concentration of 50 ppb. To our limited knowledge, 50 ppb is the minimum acetone concentration recorded to date by a sensor in an environment with a temperature below 200 °C. Furthermore, as seen from Figure 8d, the Pt-doped 3DPS-based sensor possesses a wider linear range. The response of the Pt-doped 3DPS sensor linearly increases with increasing acetone concentration, while the other two sensors seem to start saturating at a concentration of up to 500 ppm. Additionally, the inset of Figure 8d shows the response of Pt-doped 3DPS, which increases sharply from 50 ppb to 2 ppm. Figure 8e shows that the response and recovery times of the Pt-doped 3DPS sensor toward 50 ppb acetone at 153 °C are 440 and 370 s, respectively. As we have previously described, the effect on the response time of a gas sensor depends not only on the performance of the sensor itself but also on the ingress and egress rates of the gas. Nonetheless, this gas response time is acceptable for the measurement of static acetone gas. Figure 8f describes the repeatability of the Pt-doped 3DPS sensor’s response to a 50 ppb concentration. One can see that after four dynamic response-recovery cycles of charging and discharging acetone gas, the sensitivity (R_a_/R_g_) reveals no degradation, indicating that the Pt-doped 3DPS sensor has a good stability for practical applications.

It is well known that selectivity is an important parameter for characterizing gas sensors. Figure 9a shows a bar graph of the response of three sensors to a variety of gases at 153 °C, and these gases include acetone, methanol, H_2_, CO, NH_3_, and CH_4_. These gases were chosen because they are gas molecules indicative of other diseases and may be found in the breath of diabetic patients. Obviously, the response of the three materials toward acetone is much higher than that toward methanol, H_2_, CO, NH_3_, and CH_4_, which proved that all three materials have a good selectivity toward acetone. The maximum response of the Pt-doped 3DPS sensor to 100 ppm acetone is 505.7, which is 8.0, 15.1, 106.3, 19.4, and 52.1 times higher than the maximum response of the Pt-doped 3DPS sensor to 100 ppm methanol, H_2_, CO, and NH_3_ and 6000 ppm CH_4_, respectively. Figure 9b highlights the excellent acetone selectivity of Pt-doped 3DPS against 1 ppm methanol, H_2_, CO, NH_3_, and CH_4_. Highly selective property of the Pt-doped 3DPS sensor is demonstrated, confirming significantly higher acetone response (R_a_/R_g_ = 37.5 at 1 ppm) while exhibiting negligible responses (R_a_/R_g_ < 5) of other gases. Our experimental results show that the Pt-doped 3DPS sensor has an exceptional selectivity to acetone. Based on the determined response, stability, and selectivity, our results reveal that the Pt-doped 3DPS sensor is a promising sensor for detecting acetone.

Table 1 lists the overall acetone sensing performances of different material sensors reported so far and the Pt-doped 3D porous SnO_2_ sensor developed in this work for a comparison. The results indicate that the 1.0 wt % Pt-doped 3D porous SnO_2_ acetone sensor has a lower working temperature (153 °C) and the highest response toward acetone compared to the other reported sensors, which clearly demonstrates the excellent acetone sensing capabilities of the 1.0 wt % Pt-doped 3D porous SnO_2_ acetone sensor.

### 3.3. Gas Sensing Mechanism

Basically, the ionization of the oxygen formed on the surface of the metallic oxides is needed [43,44], since it determines the rate and products of the reaction. When the sample temperature is below 150 °C, the O_io_ on tin oxide is mainly O_2(ad)_^−^. When the temperature range is from 150 to 200 °C, the primary chemisorbed species become O_(ad)_^−^. As the temperature increases (>200 °C), more O_ad_ converts to O_(ad)_^−^ and O_(ad)_^2−^ by accepting electrons from tin oxide. In this experiment, the best sensitivity and selectivity occur in the temperature range of 150–200 °C, and thus, the O_(ad)_^−^ ions are considered as the most likely and dominant oxygen ions to participate in the catalytic reaction. This reaction process can be expressed by the following formula:CH_3_COCH_3 (ads)_ + 8O^−^_(ads)_ → 3CO_2 (gas)_ + 3H_2_O _(gas)_ + 8e^−^(1)
As one can see from this reaction equation, an acetone molecule releases as high as 8 electrons into the conduction band of the SnO_2_ material during the adsorption and reaction with oxygen ions to form CO_2_ and H_2_O. This explains the observed decline in the resistance during the sensing reaction.

The aims to utilize hierarchical layer structure, as show in Figure 3 and Figure 4, is to maximize specific surface area of the material, providing the most adsorption sites for incoming molecules. It will allow more oxygen ions and reactive molecules to participate in surficial process, such as the adsorption, ionization and reaction. The micropores generated by post-annealing will provide more absorbed sites, and more important, supply extra channels for gas diffusion. All above efforts, concisely, are to enhance the gas reaction and improve the sensitivity of the gas sensor.

The target to doping noble metals in sensing material is to either provide more sites for more gas absorption or reduce the absorbed activate energy for readily gas absorption or reaction [45,46,47]. As illustrated in Figure 10, since the free energy of the surface of the noble metal particles is lower than that of the metal oxide, the oxygen and the acetone molecules preferentially absorb on the metal particle but not at the metallic oxide [48]. During the absorption of oxygen and the acetone gas, part of Pt still remains in a metallic state in air, which provides not only catalytic adsorption sites for oxygen and acetone but also help the activated oxygen species to be ionosorbed through trapping electrons from the metal oxide [49,50]. Thus, the molecules can first adsorb onto the oxide support and then diffuse to a catalyst particle. This suggests that the effective capture radius of a catalyst nanoparticle can be much greater than the nanoparticle’s physical radius, consequently increasing the capture of oxygen and acetone molecular absorption and accelerating the molecular reaction [51].

The effect of the temperature on the reaction is well-known, the reaction equation can be expressed as following equation:k = A exp[−E_a_/RT](2)
where k is the reaction rate, A is the pre-exponential factor, Ea is the activation energy, R is the gas constant, and T is the temperature. In principle, the reaction rate is proportional to the temperatures. The temperature provides the activation energy required for the reaction, which is not only for the catalytic reaction of the adsorbed molecules but also for the ionization of the oxygen. As the temperature continues to increase, more activation energy supplied, leading to a boost to the reaction rate. As shown in Figure 7, the sensing response increases initially with the temperature up to 153 °C, if the temperature was more than 153 °C, the sensing response began to drop. This occurs because, at a certain high temperature, 153 °C for instance, all the molecules may achieve activation energies required for the reaction. Continuing to increase the temperature does not help to drive the reaction rate up further, but fetch a rapid desorption of absorbed molecules. The decrease in the concentration of the molecules that attend the reaction resulted in downward of sensing response. In addition, the dependence of diffusion [52,53] and desorption [54,55] on the temperature may be considered as another two-important factor affecting the performance of the sensors. Concisely, there exist an optimal temperature which can maximizes the reaction rate, says 153 °C in this work.

As shown in Figure 9, a gas sensor based on the Pt-doped 3DPS material exhibits a high selectivity toward acetone. However, the mechanism of selective adsorption and reaction of gases with sensitive materials is currently unclear. Many simulation studies have shown that because the crystal orientation of the material has different free energies, different crystals have different adsorption energies for different gases. The acetone molecule, relative to other molecules, produces selective adsorption on the (200) crystal face. The XRD data, such as the one shown in Figure 2, shows that Pt-doped 3DPS exhibits an intense diffraction peak corresponding to the (200) crystal plane. Thus, the selective adsorption of acetone molecules on the (200) crystal plane explains the selectivity of the sensor to acetone to some extent. Furthermore, noble metallic particle promoters are particularly important to metal oxide selective catalysts, as noble metallic particle promoters increase the number of suitable donor or acceptor species involved in the electron transfer processes. Since promoters can enhance the rate of reaction of one species relative to another, these promoters can have a marked effect on the catalyst selectivity [56]. As a result, the parameters of gas sensors, such as the sensor response, rate of response, and selectivity, can be dramatically improved [57].

## 4. Conclusions

Three SnO_2_-based sensor materials with similar microstructures, including pure SnO_2_ with hierarchical microflower structures, Pt-doped 3DS, and Pt-doped 3DPS, were successfully fabricated using a hydrothermal method. The samples were successfully characterized using various techniques, which confirmed that the synthesized material is well-crystalline and possesses a tetragonal rutile crystal phase. The average size of the synthesized microflowers is approximately 2–4 μm. The X-ray photoelectron spectroscopic analysis (XPS) confirmed that the PtO_X_ dopants were incorporated into the SnO_2_ products. By carefully studying the sensitivity, selectivity, response, and recovery time of the gas sensors, it is found that Pt-doped 3DPS has a superior performance compared to pure SnO_2_ and the Pt-doped 3DS. A sensing response of 505.7 can be observed for an acetone concentration of 100 ppm at 153 °C. A detection limit down to the ppb level has been achieved. For a 50 ppb acetone gas exposure, the Pt-doped 3DPS sensor has achieved a recorded response as high as 2.1 at a temperature of 153 °C. The Pt-doped 3DPS sensor also shows a very good selectivity and stability as well. The studies regarding the sensing mechanisms show that the porous, hierarchical, and dopant effects of the Pt-doped 3DPS sensor occur simultaneously, which plays a synergetic role in the observed results. The measured results demonstrated that the Pt-doped 3D porous SnO_2_ architectures can be used for the detection of low acetone concentrations and have potential for the early diagnosis of diabetes.

## Figures and Tables

**Figure 1 sensors-20-01150-f001:**
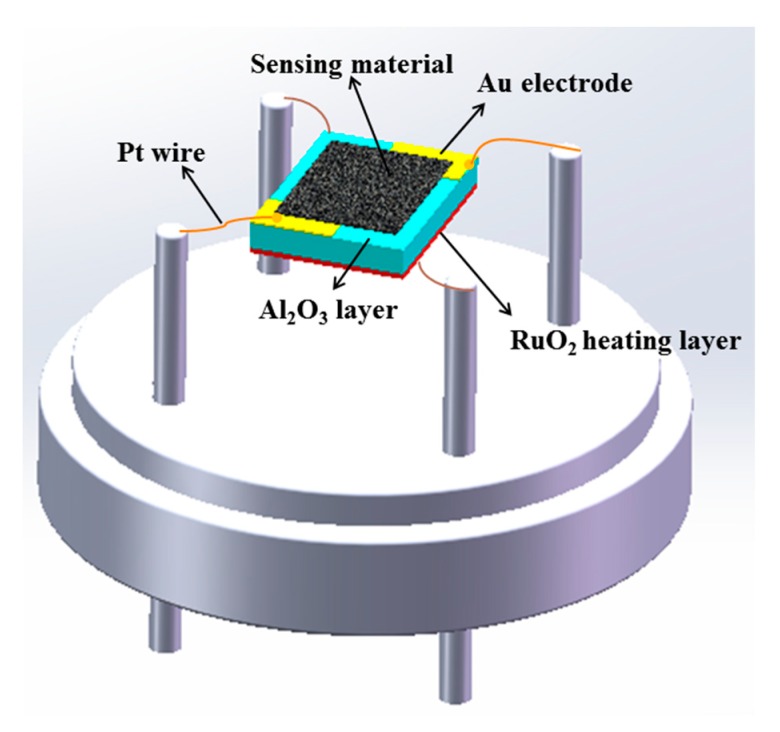
Schematic diagram of the gas sensor.

**Figure 2 sensors-20-01150-f002:**
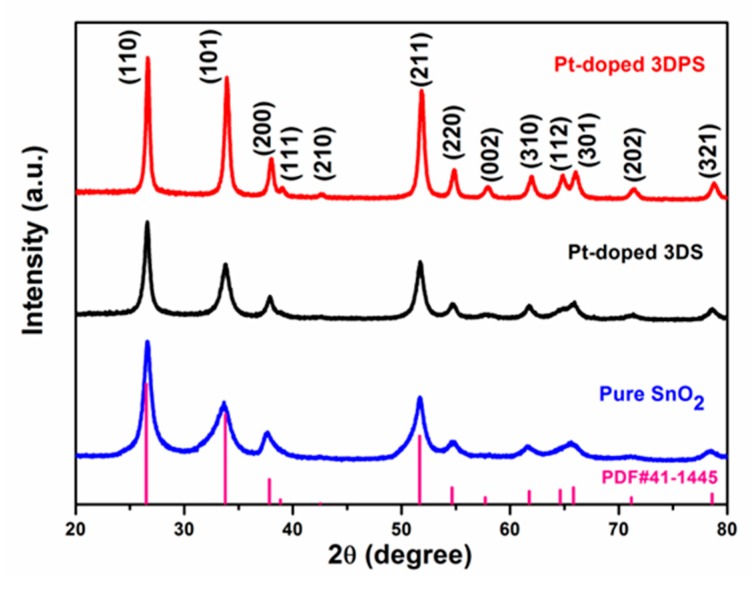
X-ray diffraction patterns of pure SnO_2_, Pt-doped 3DS and Pt-doped 3DPS.

**Figure 3 sensors-20-01150-f003:**
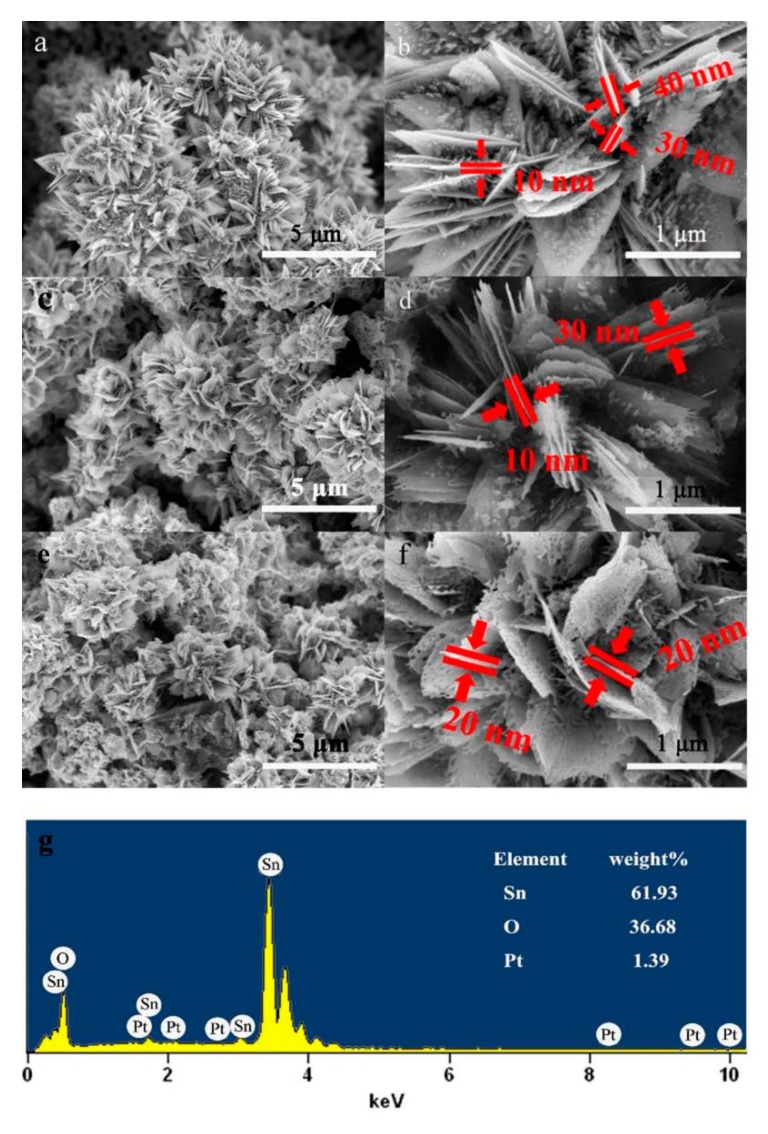
SEM micrographs of (**a**,**b**) pure SnO_2_, (**c**,**d**) Pt-doped 3DS, and (**e**,**f**) Pt-doped 3DPS and (**g**) the energy dispersive X-ray (EDX) stoichiometric analysis of Pt-doped 3DPS.

**Figure 4 sensors-20-01150-f004:**
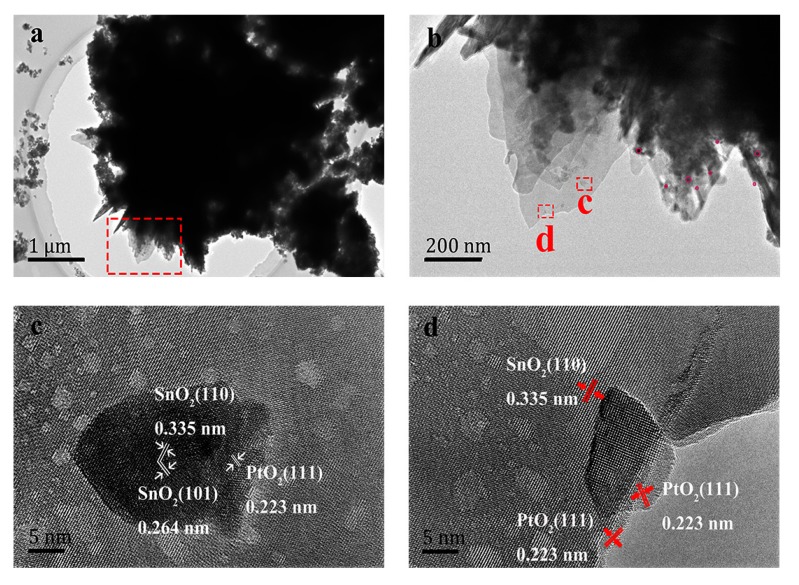
TEM (**a**,**b**) and HRTEM (**c**,**d**) micrographs of Pt-doped 3DPS.

**Figure 5 sensors-20-01150-f005:**
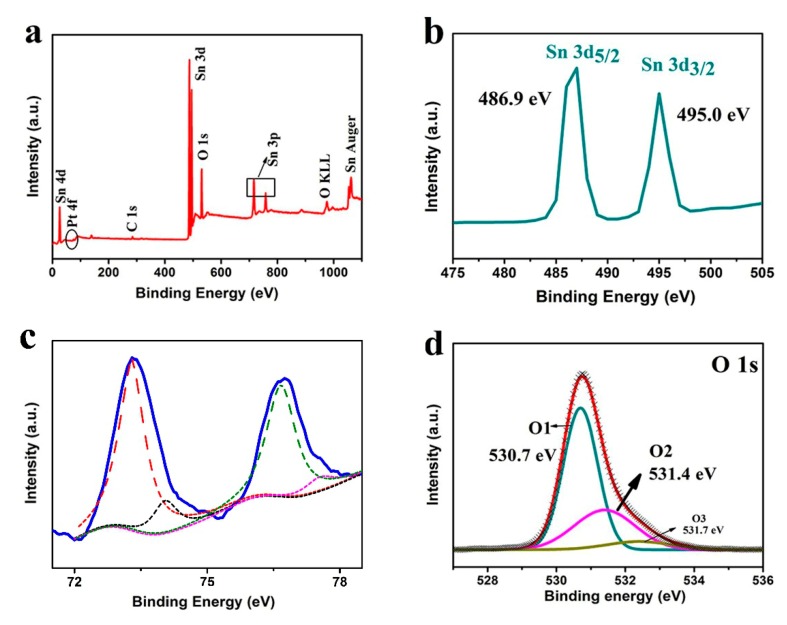
XPS spectra of (**a**) the fully scanned spectrum, (**b**) Sn 3d, (**c**)Pt 4f and (**d**) O 1s electron binding energies of Pt-doped 3DPS.

**Figure 6 sensors-20-01150-f006:**
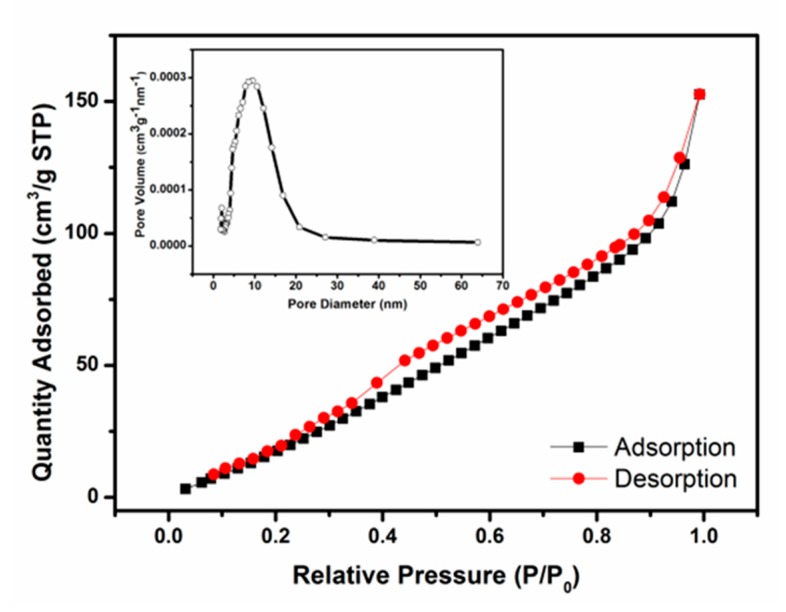
Nitrogen adsorption-desorption isotherms of Pt-doped 3DPS.

**Figure 7 sensors-20-01150-f007:**
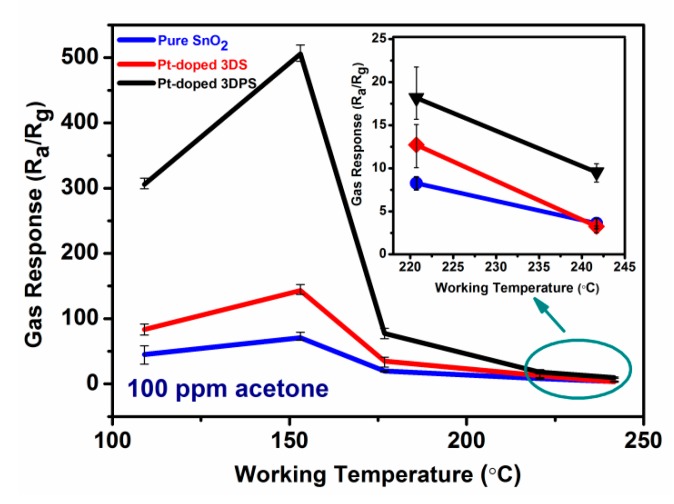
Responses of pure SnO_2_, Pt-doped 3DS and Pt-doped 3DPS to 100 ppm acetone gas at different working temperatures.

**Figure 8 sensors-20-01150-f008:**
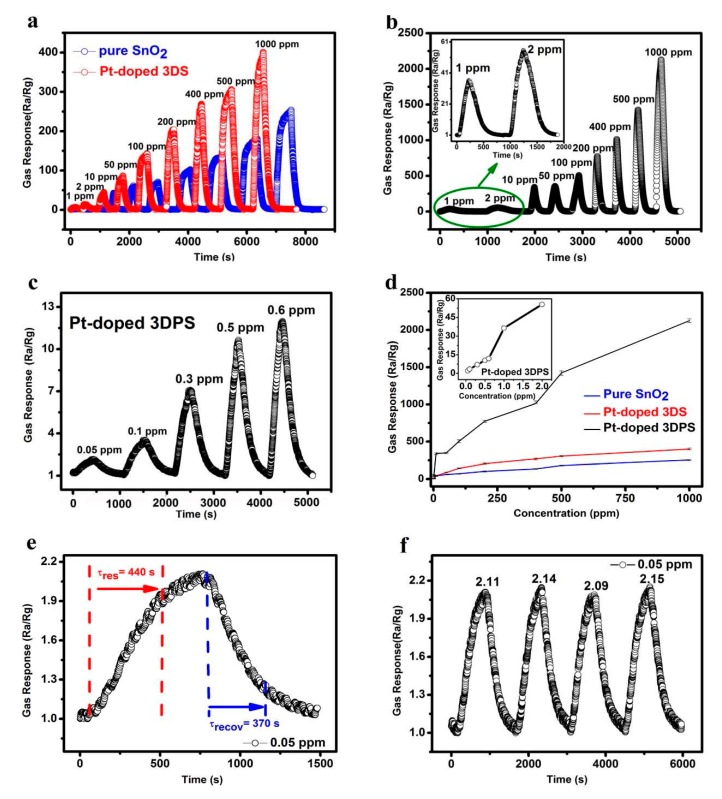
(**a**) Dynamic gas responses as a function of time to different acetone concentrations from 1 ppm to 1000 ppm at 153 °C for pure SnO_2_ and Pt-doped 3DS; (**b**) Same graph as shown in (**a**) but with the dynamic responses of Pt-doped 3DPS; (**c**) Representative dynamic response of the Pt-doped 3DPS sensor to low acetone concentrations (50 ppb-0.6 ppm) at 153 °C; (**d**) Responses as a function of the gas concentration to different acetone concentrations from 1 ppm to 1000 ppm at 153 °C for pure SnO_2_, Pt-doped 3DS and Pt-doped 3DPS, and the inset shows the responses of the Pt-doped 3DPS sensors as a function of acetone concentration for concentrations ranging from 50 ppb-2 ppm at 153 °C; (**e**) Response and recovery times of the Pt-doped 3DPS sensor to 50 ppb acetone at 153 °C; (**f**) Repeatability of the sensor based on Pt-doped 3DPS after four-cycles of exposure to 50 ppb acetone at 153 °C.

**Figure 9 sensors-20-01150-f009:**
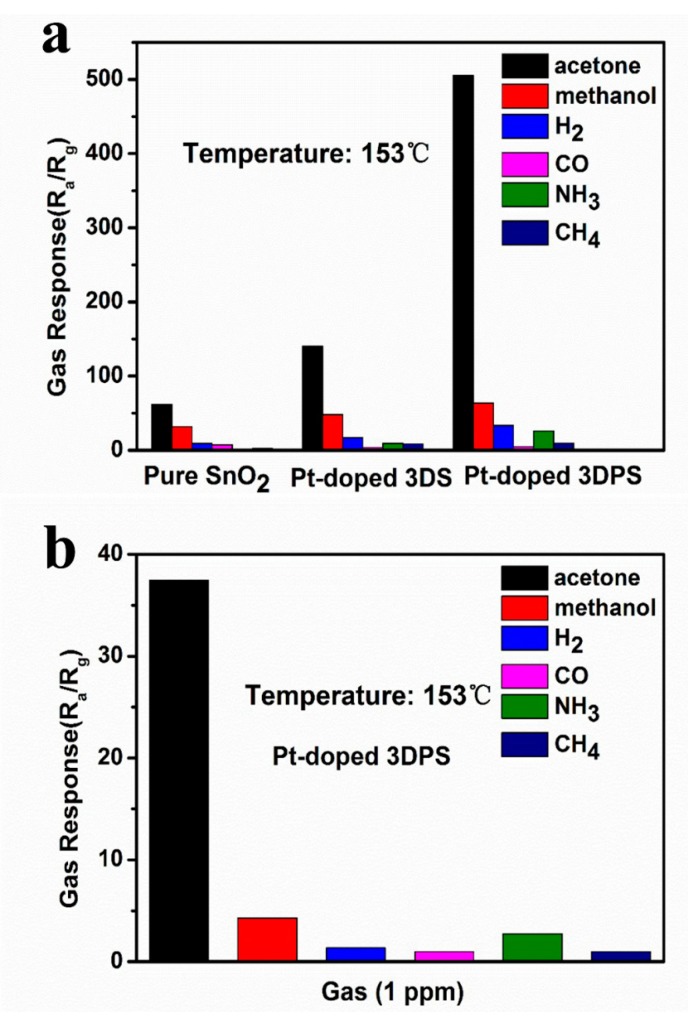
(**a**) Selective characterization of different sensors based on pure SnO_2_, Pt-doped 3DS and Pt-doped 3DPS with exposure to various gases at 153 °C; (**b**) Selective characterization of Pt-doped 3DPS at gases concentration of 1 ppm.

**Figure 10 sensors-20-01150-f010:**
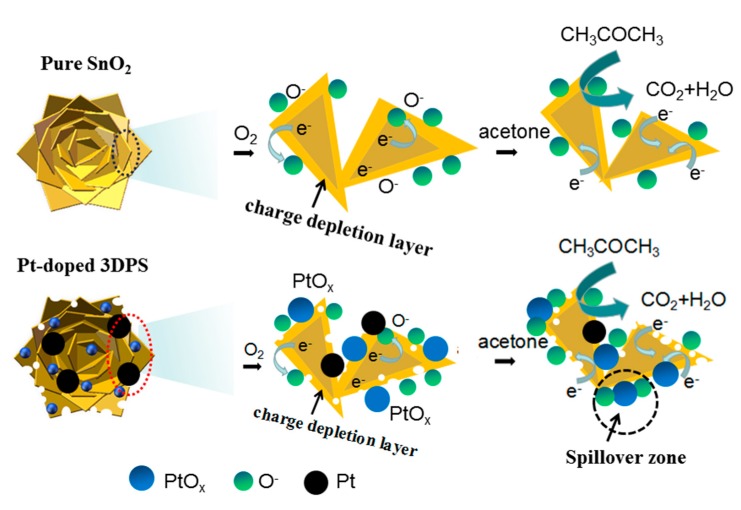
A possible gas-sensing mechanism of pure SnO_2_ and Pt-doped 3DPS hierarchical nanoflowers in ambient air and ambient acetone, respectively.

**Table 1 sensors-20-01150-t001:** Summary of the acetone gas sensing performances of different sensing materials and the Pt-doped 3DPS material developed in this work.

S.N.	Materials	Temp. (°C)	Response (Concentration)	t_res_ (s)	t_recov_ (s)	Ref.
1	Co_3_O_4_/FGH composites	250	4.06 (1 ppm)	20	---	[34]
2	Hierarchical flower-like Co_3_O_4_ nanostructures	130	48.1 (100 ppm)	18	13	[35]
3	Porous NiFe_2_O_4_ microspheres	250	27.4 (100 ppm)	2	406	[36]
4	Rh-doped SnO_2_ nano-fibers	200	60.6 (50 ppm)	2	64	[37]
5	SnO_2_/Au-doped In_2_O_3_ core-shell nano-fibers	280	14 (100 ppm)	2	9	[38]
6	Pt-PS_SnO_2_ NTs	350	34.8 (1 ppm)	---	---	[39]
7	PdAu/SnO_2_ 3D nanosheets	250	6.5 (2 ppm)	5	4	[40]
8	Pt-loaded Fe_2_O_3_ nanocubes	139	25.7 (100 ppm) and 1.1 (1 ppm)	3	22	[41]
9	Ni_1_Sn_3_	300	2.44 (0.5 ppm)	---	---	[42]
10	Pt-doped 3D porous SnO_2_	153	505.7 (100 ppm) 2.1 (0.05 ppm)	130 440	140 370	In this work

--- not mentioned.

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
