# Peer review of "A Highly Sensitive and Selective ppb-Level Acetone Sensor Based on a Pt-Doped 3D Porous SnO2 Hierarchical Structure"

_sensors, 2020, doi:10.3390/s20041150_

Round 1
Reviewer 1 Report
The manuscript is dedicated to the creation of the sensor for the acetone vapors. The paper is well written and research has been done on the high methodological level. Nevertheless, I have serious doubts about novelty and application potential for the provided sensor. Many results can be found in literature dedicated to the acetone sensing (for instance, DOI: 10.1021/acsami.7b16258, where the SnO2 in combination with PtO2 has been used). In addition, the application of relatively high working temperature can be considered as a negative factor for the sensor application.
Nevertheless, the proposed manuscript can be published in Sensors after the minor revision.
The size distribution of Pt/PtO2 nanoparticles should be given at least in SI. The peaks of Pt on the XPS spectra are slightly unsymmetrical. The deconvolution of Pt peaks and related discussion should be provided. The mechanistic insight is pretty exciting and sound. However, how authors can explain the high selectivity of the sensor towards acetone? Methanol can be oxidized in these conditions much faster.
Author Response
Reviewer 1: (Minor)
The manuscript is dedicated to the creation of the sensor for the acetone vapors. The paper is well written and research has been done on the high methodological level. Nevertheless, I have serious doubts about novelty and application potential for the provided sensor. Many results can be found in literature dedicated to the acetone sensing (for instance, DOI: 10.1021/acsami.7b16258, where the SnO2 in combination with PtO2 has been used). In addition, the application of relatively high working temperature can be considered as a negative factor for the sensor application.
Answer: thanks for the reviewer’s positive comments.
Nevertheless, the proposed manuscript can be published in Sensors after the minor revision.
The size distribution of Pt/PtO2 nanoparticles should be given at least in SI.
Answer: Thank you for your reminder. We have added the size distribution of nanoparticles in figure S1. And the discussion about it has been added in the modified manuscript as follows:
Figure S1. The size distribution of nanoparticles.
According the size distribution diagram (Figure S1), the average size of nanoparticles is estimated to be about 8.7 nm. (line 240-241)
The peaks of Pt on the XPS spectra are slightly unsymmetrical. The deconvolution of Pt peaks and related discussion should be provided.
Answer: We have added the deconvolution of Pt peaks in Figure 5c. And the discussion about it has been added in the modified manuscript as follows (line 261-265):
Figure 5c. XPS spectra of Pt 4f electron binding energies of Pt-doped 3DPS.
The Pt 4f (Figure 5c) signals display two pairs of doublets, indicating the presence of two oxidation states. The most intense doublet with binding energy of 73.3 eV (Pt 4f7/2) and 76.6 eV (Pt 4f5/2) is attributed to Pt2+ as in PtO. The second doublet found at 74.1 and 77.6 eV appears to be Pt4+, possibly as PtO2 [28-30]. In contrast, metallic Pt peaks (70-71 eV) were not detected in the crystal structure of Pt-doped 3DPS in Figure 5c[31].
[28]Hernández-Fernández, P.; Nuño, R.; Fatás, E.; Fierro, J. L. G.; et al. MWCNT-supported PtRu catalysts for the electrooxidation of methanol: effect of the functionalized support. Int. J. Hydrogen Energy 2011, 36, 8267-8278.
[29]Yasuda, K.; Nobu, M.; Masui, T.; Imanaka, N.; et al. Complete oxidation of acetaldehyde on Pt/CeO2–ZrO2–Bi2O3 catalysts. Mater. Res. Bull. 2010, 45, 1278-1282.
[30] Zheng, Y.; Qiao, J.; Yuan, J.; Shen, J.; et al. Controllable synthesis of PtPd nanocubes on graphene as advanced catalysts for ethanol oxidation. Int. J. Hydrogen Energy 2018, 43, 4902-4911.
[31]Romanovskaya, V.; Ivanovskaya, M.; Bogdanov, P. A study of sensing properties of Pt-and Au-loaded In2O3 ceramics. Sens Actuators, B 1999, 56, 31-36.
The mechanistic insight is pretty exciting and sound. However, how authors can explain the high selectivity of the sensor towards acetone? Methanol can be oxidized in these conditions much faster.
Answer: We should honestly say that achieving the selective response and theoretical explanation of a gas sensor for a certain gas is an extremely difficulty problem for metal oxide semiconductor gas sensors. The possible response of the material to the selection of acetone in this experiment are as follows: (1) Acetone is preferably absorbed on some crystal planes of the material. The use of salicylic acid in synthesis may easily control the crystal plane growth, especially promote the growth of the (200) plane. In our other submitted articles, the simulations indeed show that (200) has selective adsorption in the face of acetone molecules; (2) it may be the response of the incorporation of precious metal particles doping, which reduces the acetone adsorption energy and the activation energy of the redox reaction; (3) It may come from the fact that the current working temperature is happen and well-suitable for the adsorption and reaction of acetone, thereby enhancing the response of acetone.

Reviewer 2 Report
The submitted manuscript “A highly sensitive and selective ppb-level acetone sensor based on a Pt-doped 3D porous SnO2 hierarchical structure” reports the results of a study on the structural properties and electrical performance of 3 sensors that use hierarchical flower like SnO2 materials: a Pt-free, a Pt- containing and a calcined Pt-containing samples. The application of such sensors in the context of diabetes diagnosis is of high relevance, however a major revision is recommended based on the following questions and comments, as to improve the quality of the manuscript:
1. In the methodology, include the values of the flow rates used, as it was stated that the ingress and egress rates of gases affect the performance of sensors (lines 340-341).
2. Throughout the text, it was stated that the samples were doped with Pt, however XPS (in accordance to HR-TEM) results showed that Pt oxides (as segregated phases) were present in samples, so that Pt as a dopant (a point defect) was not present (there were also no Pt clusters detected according to XPS). The prepared samples seem to be composites of Pt oxides with SnO2.
Include a label for the vertical axis in figure 3g.
Regarding XRD results the Pt content (and thus, the Pt oxides content) is very likely to be under the detection limit of the technique: it is not the size of the particles nor their dispersion what causes the absence of peaks related to the Pt oxides phases (lines 201 and 202).
3. Do the error propagation for all the numerical results, so that every numerical result be reported with its uncertainty/error. Whenever possible, include error bars in graphs and it would be better not to join points of graphs; if needed, include trend lines.
4. If there were traces of salicylic acid in samples that were not calcined, it would have been good to test a calcined sample (Pt free) as to analyze whether those traces had an effect on the sensing response.
5. It would be good to include the surface area (BET results) of the 3 different materials, and analyze whether responses could be correlated to the differences in surface area.
6. It was stated that “153°C is the optimum working temperature”. I recommend to add “from the tested temperature values”.
7. To me, the growth mechanism of the 3D Porous SnO2 structures is not the main interest of the presented study, so that such information could be included as supplementary material.
8. In Figure 8, could there be included a representation of the pulses in the system? as to clearly see if there is a time delay in the responses. A comment on the response and recovery times compared to values from other studies, for application, (table 1) is missing. Would those times be a function of temperature? Include in table 1 results (values of response and response and recovery times) for 1 ppm acetone.
9. In lines 124-125 it was stated that the mechanism leading to high sensitivity, selectivity and optimum operating temperature was studied and explained, however there was no development of a model nor a selection of one that could be validated with experimental data of the prepared sensors, as to affirm that a mechanism was studied and it would explain the obtained results. Generalities and some results of other studies on diffusion, adsorption-desorption and surface Red-Ox reactions were included in the discussion. Furthermore, some concepts should be revised:
-In equation 5, it would be better to state whether acetone, CO2 and H2O are in the gas phase or adbsorbed, in accordance to the given explanation.
- It was stated that the micropores locally change the electronic properties of the metal oxide material (line 435). How would a pore change the electronic properties of the material? Would micropore creation during calcination involve the creation of vacancies (e.g. oxygen vacancies) or other type of point defects that could contribute to a change of intrinsic electronic properties? Changing the porosity would only alter the surface area.
-Regarding equation 6, “k” would be a reaction rate only if it would be concentration independent (i.e. zero order reaction rate), otherwise “k” is a reaction rate constant or coefficient.
- In line 500, it was stated that temperature provides a greater activation energy, however an activation energy is an energy barrier for the reaction to happens and in equation 6 it is a constant value for a specific reaction, thus the indicated expression is not correct.
- What are the expected active sites for reaction in the SnO2 and Pt-oxides phases?
-About equation 7, “K” was not defined in the text and please revise the sentence “D is the reaction rate and diffusion efficiency” (line 514); D can not represent those two processes, D is a diffusion coefficient.
10. Are the fabricated devices reproducible?
11. Regarding the interference of other gases (different to acetone, e.g. methanol, H2, CO, NH3, CH3) to the sensor´s response, there were some electrical measurements performed, but a justification is missing on the concentration values of the other gases. It would be good to mention the typical concentration range values of those gases, for the application of the sensors. If a mixture of those gases were used, would the response of the sensor be additive?
12. Some minor (mainly typographical) errors were found and marked throughout the manuscript (see attached file). Revise for figures that show a set of graphs, that the size of the text in axes labels be the same.

Author Response
Reviewer 2: (Major)
The submitted manuscript “A highly sensitive and selective ppb-level acetone sensor based on a Pt-doped 3D porous SnO2 hierarchical structure” reports the results of a study on the structural properties and electrical performance of 3 sensors that use hierarchical flower like SnO2 materials: a Pt-free, a Pt- containing and a calcined Pt-containing samples. The application of such sensors in the context of diabetes diagnosis is of high relevance, however a major revision is recommended based on the following questions and comments, as to improve the quality of the manuscript:
Answer:Many thank for the reviewer’s comments and suggestions.
In the methodology, include the values of the flow rates used, as it was stated that the ingress and egress rates of gases affect the performance of sensors (lines 340-341).
Answer: Thanks to the reviewers for their comments. In all the processes, we use the speed of the two gases to adjust the gas concentration. If the working stability of the mass flow meter is considered, these effects on the sensor performance can be neglected.
Throughout the text, it was stated that the samples were doped with Pt, however XPS (in accordance to HR-TEM) results showed that Pt oxides (as segregated phases) were present in samples, so that Pt as a dopant (a point defect) was not present (there were also no Pt clusters detected according to XPS). The prepared samples seem to be composites of Pt oxides with SnO2. Include a label for the vertical axis in figure 3g. Regarding XRD results the Pt content (and thus, the Pt oxides content) is very likely to be under the detection limit of the technique: it is not the size of the particles nor their dispersion what causes the absence of peaks related to the Pt oxides phases (lines 201 and 202).
Answer: We understand the reviewers' comments. Originally, it is Pt doping added material. During the annealing process, almost all Pt is oxidized to PtOx. PtOx also has good catalytic properties and has no effect on the results of this article.
Do the error propagation for all the numerical results, so that every numerical result be reported with its uncertainty/error. Whenever possible, include error bars in graphs and it would be better not to join points of graphs; if needed, include trend lines.
Answer: It is a very good question, thanks. Error bars added in figure 7 (line 319) and 8d (line 350), accordingly.
Figure 7. Responses of pure SnO2, Pt-doped 3DS and Pt-doped 3DPS to 100 ppm acetone gas at different working temperatures.
Figure 8d. Responses as a function of the gas concentration to different acetone concentrations from 1 ppm to 1000 ppm at 153°C for pure SnO2, Pt-doped 3DS and Pt-doped 3DPS, and the inset shows the responses of the Pt-doped 3DPS sensors as a function of acetone concentration for concentrations ranging from 50 ppb-2 ppm at 153°C.
If there were traces of salicylic acid in samples that were not calcined, it would have been good to test a calcined sample (Pt free) as to analyze whether those traces had an effect on the sensing response.
Answer: To ensure stability, all sensors have been subjected to a one-week aging treatment in air at approximately 250° C. We believe the sample is free from salicylic acid, as salicylic acid is reported to decompose below 250° C [1]. Even some salicylic acid residue that will not affect the conclusion of this article as a whole.
[1] Iijima, T.; Yamaguchi, T. Efficient regioselective carboxylation of phenol to salicylic acid with supercritical CO2 in the presence of aluminium bromide. J. Mol. Catal. A. Chem 2008, 295, 52-56.
It would be good to include the surface area (BET results) of the 3 different materials, and analyze whether responses could be correlated to the differences in surface area.
Answer: Thanks for reviewer’s suggestion. The BET results of pure SnO2 and Pt-doped 3DS materials are added in Figure S2, accordingly.
Add explanation in the article (line 289-292): As shown in Figure. S2, the pure SnO2 and Pt-doped 3DS samples possess specific surface areas of 78.29 and 129.30 m2 g−1, respectively. The insets of graphs show the pore size of the two samples. No peak was observed in the pore size distribution curves of pure SnO2 and Pt-doped 3DS samples.
Figure S2. Nitrogen adsorption-desorption isotherms of pure SnO2 and Pt-doped 3DS.
It was stated that “153°C is the optimum working temperature”. I recommend to add “from the tested temperature values”.
Answer: Agreed. We add “from the tested temperature values” at line 311.
To me, the growth mechanism of the 3D Porous SnO2 structures is not the main interest of the presented study, so that such information could be included as supplementary material.
Answer: Thank you for your suggestion. By following your suggestion, we have put the growth mechanism into the supplementary material.
In Figure 8, could there be included a representation of the pulses in the system? As to clearly see if there is a time delay in the responses. A comment on the response and recovery times compared to values from other studies, for application, (table 1) is missing. Would those times be a function of temperature? Include in table 1 results (values of response and response and recovery times) for 1 ppm acetone.
Answer: Reviewers put forward valid points. However, in an 18-litre static test system, the gas needs sufficient time to reach the set concentration, so the response time of the sensor is not the real response time, but a process of gradually accumulating gas concentration. The desorption process is the same. It is also a process of decreasing gas concentration. Response time is important, but it cannot be compared with other reported results here, because the experimental equipment can be different.
There are some response and recovery times in table1 that are not mentioned in those articles, therefore we add explanation at the bottom of table1 to explain. Thank you for your reminding. We have included those results for 100 ppm acetone in the article (130 and 140 s, respectively).
In lines 124-125 it was stated that the mechanism leading to high sensitivity, selectivity and optimum operating temperature was studied and explained, however there was no development of a model nor a selection of one that could be validated with experimental data of the prepared sensors, as to affirm that a mechanism was studied and it would explain the obtained results. Generalities and some results of other studies on diffusion, adsorption-desorption and surface Red-Ox reactions were included in the discussion. Furthermore, some concepts should be revised:
-In equation 5, it would be better to state whether acetone, CO2 and H2O are in the gas phase or adsorbed, in accordance to the given explanation.
Answer: In this article, we did not process a therotical simulation to explain the observed result. The result is pretty consistent with current established theory. That is why we wrote three paragraphies to explain the result in all aspects, diffusion, adsorption-desorption and surface Red-Ox reaction. Furthermore, we revised manucstipt and define acetone, CO2 and H2O are adsorbed gases, in accordance to the given explanation.
Modify equation 5 as follows and can be seen in line 408:
CH3COCH3 (ads) + 8O-ads → 3CO2 (gas) + 3H2O(gas) + 8e-
where the ads means the state of adsorption, the gas is in the gas phase.
- It was stated that the micropores locally change the electronic properties of the metal oxide material (line 435). How would a pore change the electronic properties of the material? Would micropore creation during calcination involve the creation of vacancies (e.g. oxygen vacancies) or other type of point defects that could contribute to a change of intrinsic electronic properties? Changing the porosity would only alter the surface area.
Answer: We appreciate the comments of the reviewers. The micropores themselves cannot change the electronic properties of the material, but the annealing process that generates the micropores will cause defects or defect recombination, which can locally change the electronic properties of the metal oxide material. We made some changes, please see at line 418.
-Regarding equation 6, “k” would be a reaction rate only if it would be concentration independent (i.e. zero order reaction rate), otherwise “k” is a reaction rate constant or coefficient.
Answer: You are right, we agreed with the reviewer’s opinion.
- In line 500, it was stated that temperature provides a greater activation energy, however an activation energy is an energy barrier for the reaction to happens and in equation 6 it is a constant value for a specific reaction, thus the indicated expression is not correct.
Answer: The high temperature would change the activation energy. Ea changes with the temperature. However, our experiments were performed at a constant temperature, the changes of DEa caused by the temperature is still constant and thus Ea maintains as a constant under a certain temperature.
Modify equation 6 as follows (line 479):
k = A exp[-Ea/RT]
- What are the expected active sites for reaction in the SnO2 and Pt-oxides phases?
Answer: Yes, we agree. It is believed that it is possible to enhance oxygen adsorption and change to an adsorbed state at PtOx positions. Only by changing to adsorbed oxygen can the change in resistance be caused.
-About equation 7, “K” was not defined in the text and please revise the sentence “D is the reaction rate and diffusion efficiency” (line 514); D can not represent those two processes, D is a diffusion coefficient.
Answer: Many thanks! The phase has been revised as “here K is the reaction rate and D is diffusion efficiency”. Please see the description below equation 7 (line 500).
Are the fabricated devices reproducible?
Answer: We normally and experimentally take three samples for our performance characterizations, first one for screening to find best sample, second one for confirmation and third one for repeatability test. In spite of sampling size using in this experiment below industrial standard, it believes that the observed results are reliable and repeatable.
Regarding the interference of other gases (different to acetone, e.g. methanol, H2, CO, NH3, CH4) to the sensor´s response, there were some electrical measurements performed, but a justification is missing on the concentration values of the other gases. It would be good to mention the typical concentration range values of those gases, for the application of the sensors. If a mixture of those gases were used, would the response of the sensor be additive?
Answer: Thank you for the reviewer's question. We did measure sensor response with other gases (acetone, such as methanol, H2, CO, NH3, CH4). Selective test gas concentrations are: methanol, H2, CO, and NH3, all 100 ppm and CH4 6000 ppm. Our goal is to compare the response of acetone and other gases at the same concentration to determine the selectivity of the test for acetone. The response concentration of CH4 is relatively high, so 6000ppm was selected. Selection of these measured gas concentrations and other gas responses are consistent with other reports in the literature, such as H2 concentration range 0.5-2000 ppm [1], CO concentration range 1-2000 ppm [2, 3], and NH3 concentration range 1-1000 ppm [4] The CH4 concentration range is 250-10000 ppm [5], and the methanol concentration range is 0.1-5000 ppm [6].In addition, if a mixture of those gases were used, the response of the sensor will increase.
[1]Motaung, D.E.; Mhlongo, G.H.; Makgwane, P.R.; et al. Ultra-high sensitive and selective H2 gas sensor manifested by interface of n–n heterostructure of CeO2-SnO2 nanoparticles. Sens Actuators, B 2018, 254, 984-995.
[2]Wang, Q.; Wang, C.; Sun, H.; et al. Microwave assisted synthesis of hierarchical Pd/SnO2 nanostructures for CO gas sensor. Sens Actuators, B 2016, 222, 257-263.
[3]Jamnani, S,R.; Moghaddam, H.M.; Leonardi, S.G.; et al. Synthesis and characterization of Sm2O3 nanorods for application as a novel CO gas sensor. Appl. Surf. Sci 2019, 487, 793-800.
[4]Chen, H.I.; Hsiao, C.Y.; Chen, W.C.; et al. Characteristics of a Pt/NiO thin film-based ammonia gas sensor. Sens Actuators, B 2018, 256, 962-967.
[5]Bunpang, K.; Wisitsoraat, A.; Tuantranont, A.; et al. Highly selective and sensitive CH4 gas sensors based on flame-spray-made Cr-doped SnO2 particulate films. Sens Actuators, B 2019, 291, 177-191.
[6]Li, Y.; Deng, D.; Xing, X.; et al. A high performance methanol gas sensor based on palladium-platinum-In2O3 composited nanocrystalline SnO2. Sens Actuators, B 2016, 237, 133-141.
Some minor (mainly typographical) errors were found and marked throughout the manuscript (see attached file). Revise for figures that show a set of graphs, that the size of the text in axes labels be the same.
Answer: Corrected (line 163, 168, 194, 211, 320, 345, 350, 519, 645) and thanks.

Reviewer 3 Report
The paper is well written, the morphological analysis is excellent.
The results are impressive for the Pt doped samples but I was left with several questions, of which some are standard for SnO2 sensor studies.
1 The 153C ideal temperature should have been supported with more temperature points around that temperature. This would have helped in the adsorption/desorption discussion if the decline in sensitivity with increased temperature was better quantified.
2 I worry about the baseline in figures 8a,b and c,f: they are artificially flat due to the y axis units, not allowing us to see the change in the baseline resistance after each gassing cycle. this is important with n-type chemresistors.
3 Response time was shown in only one graph and it was 440 and 370 seconds, significantly slower than listed work from other authors. This should have been discussed, especially since the favoured application is breath analysis where response time should be fast.
4 Acetone was compared with other gases without similar molecular structures. Why not also test MEK, aldehydes, other ketones, carboxylic acids for selectivity? I concluded that anything with C=O would perform the same.
5 SnO2 normally has very high humidity sensitivity. this was not tested, but with breath analysis as a favoured application, RH is very important. The authors should run and report on RH sensitivity both at 50 to 250 ppb acetone and in zero air.
Small point: A very well written paper, but lines 446 to 448: "shown", not show and please rewrite the sentence for clarity.
Author Response
Reviewer 3:
The paper is well written, the morphological analysis is excellent.
The results are impressive for the Pt doped samples but I was left with several questions, of which some are standard for SnO2 sensor studies.
We thank for the reviewer encouraging comments.
The 153°C ideal temperature should have been supported with more temperature points around that temperature. This would have helped in the adsorption/desorption discussion if the decline in sensitivity with increased temperature was better quantified.
Answer: As shown in Working principle of the gas sensing test system (Figure 1), the load resistance (RL) is connected in series to the sensor. The output signal voltage (Vout) was collected by a computer at a test circuit voltage (Vc) of 5 V and a certain heating voltage. The heating voltage represents the operating temperature of the sensor. In this experiment, the heating voltage range is 3 V-5 V, with a gradient of 0.5 V. The heating voltage of 3, 3.5, 4, 4.5 and 5 V respectively corresponds to the working temperature of 109, 153, 176, 220 and 240°C. The suggestions to add more temperature points is appreciated but hardly achieved in using current apparatus.
Figure 1. Working principle of the gas sensing test system.
I worry about the baseline in figures 8a, b and c, f: they are artificially flat due to the y axis units, not allowing us to see the change in the baseline resistance after each gassing cycle. This is important with n-type chemresistors.
Answer: Thanks for your comments. The scale down time bar would illusion such a question. As shown in the figure 2, you can clearly see the change of baseline value by appropriately enlarging the baseline of figure 8a, b and c appropriately.
Figure 2. (a) Dynamic gas responses as a function of time to different acetone concentrations from 1 ppm to 1000 ppm at 153°C for pure SnO2 and Pt-doped 3DS. (b) Same graph as shown in (a) but with the dynamic responses of Pt-doped 3DPS. (c) Representative dynamic response of the Pt-doped 3DPS sensor to low acetone concentrations (50 ppb-0.6 ppm) at 153°C.
Response time was shown in only one graph and it was 440 and 370 seconds, significantly slower than listed work from other authors. This should have been discussed, especially since the favored application is breath analysis where response time should be fast.
Answer: We appreciated the reviewer’s response. Response time fully depends on the nature of the material and the test system. In a static test system, the gas needs sufficient time to reach the set concentration. At the same time, it expected to take a long time for the porous material to adsorb up to saturation. We are not worried about the effect of response time on the actual early diagnosis of the disease by measuring the breathing gas, because a static test system will be used to respond to longer adsorption times, while a heating or optically-assisted desorption system is implemented to accelerate the desorption process.
Acetone was compared with other gases without similar molecular structures. Why not also test MEK, aldehydes, other ketones, carboxylic acids for selectivity? I concluded that anything with C=O would perform the same.
Answer: According to previous reports, methanol, CO, NH3, CH4, and H2 are biomarkers of obesity, heart disease, asthma, kidney disease, and irritable bowel syndrome, which may be the interference gases for diabetes diagnosis[1-6]. To distinguish those gases with acetone gas can enhance the accuracy of diabetes diagnosis
[1] Wang, L.; Kalyanasundaram, K.; Stanacevic, M.; et al. Nanosensor device for breath acetone detection. Sensor Lett. 2010, 8, 709-712.
[2] Jang, J.S.; Choi, S.J.; Kim, S.J.; et al. Rational Design of highly porous SnO2 nanotubes functionalized with biomimetic nanocatalysts for direct observation of simulated diabetes. Adv. Funct. Mater. 2016, 26, 4740-4748.
[3] Washio, J.; Sato, T.; Koseki, T.; et al. Hydrogen sulfide-producing bacteria in tongue biofilm and their relationship with oral malodour. J. Med. Microbiol. 2005, 54, 889-895.
[4] Choi, S.J.; Kim, S.J.; Koo, W.T.; et al. Catalyst-loaded porous WO3 nanofibers using catalyst-decorated polystyrene colloid templates for detection of biomarker molecules. Chem. Commun. 2015, 51, 2609-2612.
[5] de Lacy Costello, B.P.J.; Ledochowski, M.; Ratcliffe, N.M. The importance of methane breath testing: a review. J. Breath Res. 2013, 7, 024001.
[6] Smith, D.; Turner, C.; ŠpanÄ›l, P. Volatile metabolites in the exhaled breath of healthy volunteers: their levels and distributions. J. Breath Res. 2007, 1, 014004.
SnO2 normally has very high humidity sensitivity. This was not tested, but with breath analysis as a favoured application, RH is very important. The authors should run and report on RH sensitivity both at 50 to 250 ppb acetone and in zero air.
Answer: Thanks for the reviewers for their suggestions. Humidity has a very important effect on the performance of metal oxide-based gas sensors. Research in this area has been very deep and endless. Numerous theoretical and experimental data have been established. Therefore, measurement and borrowing those theory and data should be considered in subsequent optimization experiments.
Small point: A very well written paper, but lines 446 to 448: "shown", not show and please rewrite the sentence for clarity.
Answer: Corrected and thanks.

Reviewer 4 Report
In principle, this article presents a technical report on the results of testing a sensor, albeit with high sensitivity, manufactured using a certain technology. This article does not provide any new knowledge, since the authors did not even try to understand the nature of the phenomena responsible for the observed effects. I have a negative attitude to such articles, since the authors themselves do not understand why their sensors have such parameters.
It is created the feeling that the authors decided to include in the article everything that they know about the mechanism of gas sensitivity of metal oxides, although they know very little. Moreover, they describe the generally accepted understanding of the processes involved in gas sensitivity, which has already been described in hundreds of already published articles and books. Therefore, the authors do not give any new understanding. In general, I believe that the article should be reduced by 3 times. Therefore, I propose to remove or minimize all of these unnecessary arguments from the article (3.3 Growth of Pt-doped 3D Porous SnO2 and 3.4 Gas Sensing Mechanism).
The measurement technique used by the authors raises many questions:
Why the stabilization (high-temperature annealing) was performed only for the Pt-doped 3DPS sensors? Lack of stabilization is a source of instability in sensor characteristics. In addition, without annealing, there is a high probability of the presence of hydroxides with physicochemical properties different from those of oxides. How was the temperature controlled with such accuracy? What was the source of acetone vapor? How was the gas mixture formed? It is known that at low temperatures humidity has a significant effect on sensors characteristics. However, this effect has not been evaluated and the constancy of humidity during the measurement process is not ensured. This is especially important for sensors designed to analyze the composition of exhaled air (the breathing gas). How was the increase in sensitivity of XPS measurements achieved when measuring the XPS spectra of Pt? Why the Pt concentration was not estimated by XPS? Why describe in detail results what is not used in the analysis of the main results? What is the thickness of the sensitive layer? What are the temperature conditions for manufacturing sensors?
Many questions to discuss the results.
When you look at the data provided by the authors for a sensor response with an accuracy of tenths such as 70.9, 143.1 or 505.7, then you immediately become convinced that the authors measured once on three different samples and therefore they have no data regarding the reproducibility of the results. If we repeat the process of synthesis and manufacturing of sensors, will the same results be obtained? How can we talk about the maximum sensitivity at 153°C, if the measurements were carried out in increments of 25-50°C? Moreover, apparently, the temperature is not uniform over the area of the sensor chip. The reason for such a slow sensory response is not clear? Is this related to the properties of the material or to the peculiarity of the measuring system? Otherwise, there is no need to talk about the kinetics of the sensory response. Did you control the time of the change of atmosphere in the measuring chamber? Flower-like microstructure have very high gas permeability (porosity) and therefore it is not clear what kind of pores with a diameter of 10 nm exist in structures and how do they control the adsorption-desorption processes occurring in them? There is no answer to the question - what element of the structure determines the sensitivity? It is known that in such branched structures, many elements do not affect the sensor response, since they do not participate in current transport. What is the role of annealing in increasing sensitivity? Why did annealing increase the sensitivity despite the fact that the surface area appears to have decreased? How do you explain such large clusters of platinum (10-20 nm) at such a low concentration (1%)?

Author Response
Reviewer 4: (Major)
Major revision.
In principle, this article presents a technical report on the results of testing a sensor, albeit with high sensitivity, manufactured using a certain technology. This article does not provide any new knowledge, since the authors did not even try to understand the nature of the phenomena responsible for the observed effects. I have a negative attitude to such articles, since the authors themselves do not understand why their sensors have such parameters.
It is created the feeling that the authors decided to include in the article everything that they know about the mechanism of gas sensitivity of metal oxides, although they know very little. Moreover, they describe the generally accepted understanding of the processes involved in gas sensitivity, which has already been described in hundreds of already published articles and books. Therefore, the authors do not give any new understanding. In general, I believe that the article should be reduced by 3 times. Therefore, I propose to remove or minimize all of these unnecessary arguments from the article (3.3 Growth of Pt-doped 3D Porous SnO2 and 3.4 Gas Sensing Mechanism).
Answer: The reviewers' opinions and comments are partially correct, thank you very much. However, the annealing process of 3DPS, Pt-doping 3DPS and Pt-doping 3DPS is a trinity, and it is indispensable to study the complete system. Although there are countless studies on the mechanism of metal oxide-based semiconductor gas sensors, so far, there is no complete theory to explain the gas response, especially the selectivity. Similarly, the effects of different gases, especially acetone, on special 3DPS, Pt-doping 3DPS and annealed Pt-doping 3DPS materials, although theoretically predictable, the experimental results are still new and interesting. We appreciate the reviewers' insights and suggestions, but feel that maintaining these sections will greatly help the integrity of the article, especially the systematic nature of the reader's reading.
The measurement technique used by the authors raises many questions:
Why the stabilization (high-temperature annealing) was performed only for the Pt-doped 3DPS sensors? Lack of stabilization is a source of instability in sensor characteristics. In addition, without annealing, there is a high probability of the presence of hydroxides with physicochemical properties different from those of oxides.
Answer: Thanks for the reviewer’s comments. All sensors were subjected to an aging process at approximately 250°C for one week in air in order to ensure stability, since salicylic acid is reported to decompose below 250°C. The main purpose of annealing is to form micropores structure to increase the number of adsorption sites and diffusion channels available for the gas and create more gas diffusion channels.
How was the temperature controlled with such accuracy?
Answer: Thanks for this question. As shown in Working principle of the gas sensing test system (Figure 1), the load resistance (RL) is connected in series to the sensor. The output signal voltage (Vout) was collected by a computer at a test circuit voltage (Vc) of 5 V and a certain heating voltage. The heating voltage represents the operating temperature of the sensor. In this experiment, the heating voltage range is 3 V-5 V, with a gradient of 0.5 V. The heating voltage of 3, 3.5, 4, 4.5 and 5 V respectively corresponds to the working temperature of 109, 153, 176, 220 and 240°C. The suggestions to add more temperature points is appreciated but hardly achieved in using current apparatus.
Figure 1. Working principle of the gas sensing test system.
What was the source of acetone vapor? How was the gas mixture formed?
Answer: There are two preparation methods for different acetone concentrations: (1) Preparation of high concentration acetone (10 ppm to 1000 ppm): acetone vapor directly comes from acetone solution. The micro syringe injects different concentrations of acetone liquid into the test chamber (250 ml volume) through a rubber stopper, and forms acetone vapor after heating. (2) Preparation of low concentration (50 ppb-2 ppm) acetone: in order to ensure the accuracy of gas concentration, two mass flow meters (MF) are used, one is acetone and the other is high air. An adjustment for the ratio of two gases flow ratio to form the getting acetone gas concentration.
It is known that at low temperatures humidity has a significant effect on sensors characteristics. However, this effect has not been evaluated and the constancy of humidity during the measurement process is not ensured. This is especially important for sensors designed to analyze the composition of exhaled air (the breathing gas).
Answer: Thanks for the reviewers for their suggestions. Humidity has a very important effect on the performance of metal oxide-based gas sensors. Research in this area has been very deep and endless. Numerous theoretical and experimental data have been established. Therefore, measurement and borrowing those theory and data should be considered in subsequent optimization experiments.
How was the increase in sensitivity of XPS measurements achieved when measuring the XPS spectra of Pt? Why the Pt concentration was not estimated by XPS?
Answer: We have added the deconvolution of Pt peaks in Figure 5c. And the discussion about it has been added in the modified manuscript as follows (line 261-265): XPS shows a Pt concentration of 0.21 at%. We added a simple description at line 272.
Figure 5c. XPS spectra of Pt 4f electron binding energies of Pt-doped 3DPS.
The Pt 4f (Figure 5c) signals display two pairs of doublets, indicating the presence of two oxidation states. The most intense doublet with binding energy of 73.3 eV (Pt 4f7/2) and 76.6 eV (Pt 4f5/2) is attributed to Pt2+ as in PtO. The second doublet found at 74.1 and 77.6 eV appears to be Pt4+, possibly as PtO2 [28-30]. In contrast, metallic Pt peaks (70-71 eV) were not detected in the crystal structure of Pt-doped 3DPS in Figure 5c[31].
[28]Hernández-Fernández, P.; Nuño, R.; Fatás, E.; Fierro, J. L. G.; et al. MWCNT-supported PtRu catalysts for the electrooxidation of methanol: effect of the functionalized support. Int. J. Hydrogen Energy 2011, 36, 8267-8278.
[29]Yasuda, K.; Nobu, M.; Masui, T.; Imanaka, N.; et al. Complete oxidation of acetaldehyde on Pt/CeO2–ZrO2–Bi2O3 catalysts. Mater. Res. Bull. 2010, 45, 1278-1282.
[30] Zheng, Y.; Qiao, J.; Yuan, J.; Shen, J.; et al. Controllable synthesis of PtPd nanocubes on graphene as advanced catalysts for ethanol oxidation. Int. J. Hydrogen Energy 2018, 43, 4902-4911.
[31]Romanovskaya, V.; Ivanovskaya, M.; Bogdanov, P. A study of sensing properties of Pt-and Au-loaded In2O3 ceramics. Sens Actuators, B 1999, 56, 31-36.
Why describe in detail results what is not used in the analysis of the main results?
Answer: Thanks for the reviewer’s insight. Although there is no new explanation in theory, the experimental data and the currently accepted theoretical analysis should be consistent. We wrote a systematic theoretical analysis to help readers better understand the working principle and process of semiconductor gas sensors based on metal oxides,especially for the effect of optimal operating temperature, nanostructures, and noble metal doping and nanoporous structure on adsorption, redox reaction and desorption and related to sensing performance.
What is the thickness of the sensitive layer?
Answer: The SnO2 powder materials were dispersed in ethanol to form a colloid, and then the colloid was coated on an alumina chip by spin-coating machine at room temperature (500 r.p.m). After one layer of spin coating, the sensor is dried in an oven at 60 °C for 3 min. Repeat the above steps until five times. The thicknesses of sensitive layer film is estimated to be about 5um.
What are the temperature conditions for manufacturing sensors?
Answer: Thanks. At room temperature, the sensitive material is coated on the surface of the electrode. And then dry the sensor in an oven at 60 °C. The sensor is loaded on the aging table and aged at 250 °C for about a week.
Many questions to discuss the results.
When you look at the data provided by the authors for a sensor response with an accuracy of tenths such as 70.9, 143.1 or 505.7, then you immediately become convinced that the authors measured once on three different samples and therefore they have no data regarding the reproducibility of the results. If we repeat the process of synthesis and manufacturing of sensors, will the same results be obtained?
Answer: The reviewer made an effective point, thanks. We usually perform repeatability tests on all samples. As shown in the figure 2 below (also shown in the figure 8f in the manuscript), four cycles of measurement, at a temperature of 153 °C and exposed to 100 ppm acetones, showed excellent repeatability.
Figure 2. Repeatability of the sensor based on Pt-doped 3DPS after five-cycles of exposure to 100 ppm acetone at 153°C.
How can we talk about the maximum sensitivity at 153°C, if the measurements were carried out in increments of 25-50°C? Moreover, apparently, the temperature is not uniform over the area of the sensor chip.
Answer: Thanks for this question. As shown in Working principle of the gas sensing test system (Figure 3), the load resistance (RL) is connected in series to the sensor. The output signal voltage (Vout) was collected by a computer at a test circuit voltage (Vc) of 5 V and a certain heating voltage. The heating voltage represents the operating temperature of the sensor. In this experiment, the heating voltage range is 3 V-5 V, with a gradient of 0.5 V. The heating voltage of 3, 3.5, 4, 4.5 and 5 V respectively corresponds to the working temperature of 109, 153, 176, 220 and 240â—¦C. The suggestions to add more temperature points is appreciated but hardly achieved in using current apparatus.
Figure 3. Working principle of the gas sensing test system.
The reason for such a slow sensory response is not clear? Is this related to the properties of the material or to the peculiarity of the measuring system? Otherwise, there is no need to talk about the kinetics of the sensory response. Did you control the time of the change of atmosphere in the measuring chamber?
Answer: The reviewer made an effective point. Response time fully depends on the nature of the material and the test system. In a static test system, the gas needs sufficient time to reach the set concentration. At the same time, it expected to take a long time for the porous material to adsorb up to saturation. That is why the response time takes long. Nevertheless, to compare response times among different materials are still meaningful, since the experimental procedures are same and response time may solely be resulting from the properties of the material. It is worth of mention that we may not worry about the effect of response time on the actual early diagnosis of the disease by measuring the breathing gas, because a heating or optically-assisted desorption system would be implemented to accelerate the desorption process.
Flower-like microstructure have very high gas permeability (porosity) and therefore it is not clear what kind of pores with a diameter of 10 nm exist in structures and how do they control the adsorption-desorption processes occurring in them?
Answer: We appreciated the reviewer’s comments. The purpose of generating micropores in the flower-like microstructure by annealing is to (1) increase the specific surface area and (2) provide a diffusion channel for the gas from the surface to the back. The purpose of these micropores is to enhance adsorption and increase the sensitivity of the gas sensor.
There is no answer to the question - what element of the structure determines the sensitivity? It is known that in such branched structures, many elements do not affect the sensor response, since they do not participate in current transport.
Answer: Thank you for the reviewer's question. Answering which element of the structure determines sensitivity is a difficult question. There are many factors that affect the sensitivity of metal oxide-based semiconductor sensors. There are some established theories. Basically, sensitivity involves the adsorption and reaction of molecules on the surface of sensitive materials. Any factor that can enhance the adsorption and reaction of molecules on the surface of the material can increase sensitivity. In this article, the purpose of synthesizing a flower-like structure is to increase the specific surface area. Pt-doping is to reduce the activation energy of adsorption and reaction to enhance adsorption and reaction. The micropores are created by the annealing process to increase specific surface area and diffusion channels. All of these efforts have improved material-to-gas pair sensitivity. Regards the elements of oxides, the adsorbed sites of preferential absorption play an important role in the sensitivity. Pt is most important site for molecular adsorption.
We agree with the statement that in such a branch structure, many elements may not affect the sensor response because they do not participate in current transmission. Although the contribution to sensitivity still comes from the surface of the materials, the key is that increasing the specific surface area can increase the opportunity for the molecules, after all, the adsorption and reaction only occurs on the surface of the material to contact.
What is the role of annealing in increasing sensitivity? Why did annealing increase the sensitivity despite the fact that the surface area appears to have decreased?
Answer: Thank you for the reviewer's question. The micropores are created by the annealing process to increase specific surface area and diffusion channels. All of these efforts have improved material-to-gas pair sensitivity.
How do you explain such large clusters of platinum (10-20 nm) at such a low concentration (1%)?
Answer: According the size distribution diagram, the average size of platinum nanoparticles is estimated to be about 8-12 nm. For the Pt-doped 3DPS material, after annealing at 700°C, extensive agglomeration of the platinum nanoparticles resulted in significant increase of particle size.

Round 2
Reviewer 2 Report
Few minor errors were found and marked in the attached file. Appreciate the fact that all my comments were taken into account.

Author Response
Thanks for reviewer’s suggestion. We have corrected errors in line 241, 275 and 401.

Reviewer 4 Report
The main comments remained the same
Author Response
Major revision.
In principle, this article presents a technical report on the results of testing a sensor, albeit with high sensitivity, manufactured using a certain technology. This article does not provide any new knowledge, since the authors did not even try to understand the nature of the phenomena responsible for the observed effects. I have a negative attitude to such articles, since the authors themselves do not understand why their sensors have such parameters.
It is created the feeling that the authors decided to include in the article everything that they know about the mechanism of gas sensitivity of metal oxides, although they know very little. Moreover, they describe the generally accepted understanding of the processes involved in gas sensitivity, which has already been described in hundreds of already published articles and books. Therefore, the authors do not give any new understanding. In general, I believe that the article should be reduced by 3 times. Therefore, I propose to remove or minimize all of these unnecessary arguments from the article (3.3 Growth of Pt-doped 3D Porous SnO2 and 3.4 Gas Sensing Mechanism).
Answer: The reviewer made an effective point, thanks. According to the reviewer's request, we delete the most of gossiping mechanism analysis and discussion, please see 3.3 Gas Sensing Mechanism. In addition, we put the growth mechanism into the support information. The annealing process of 3DPS, Pt-doping 3DPS and Pt-doping 3DPS is a trinity and indispensable to study the complete system. Similarly, the effects of different gases, especially acetone, on special 3DPS, Pt-doping 3DPS and annealed Pt-doping 3DPS materials, although theoretically predictable, the experimental results are still new and interesting. While we appreciated the reviewers' insights and suggestions, but still feel that maintaining this section will greatly help to keep the integrity of the article.
The measurement technique used by the authors raises many questions:
Why the stabilization (high-temperature annealing) was performed only for the Pt-doped 3DPS sensors? Lack of stabilization is a source of instability in sensor characteristics. In addition, without annealing, there is a high probability of the presence of hydroxides with physicochemical properties different from those of oxides.
Answer: Thanks for the reviewer’s comments.
Q: Why the stabilization (high-temperature annealing) was performed only for the Pt-doped 3DPS sensors?
Answer: All sensors not only Pt-doped 3DPS sensors were subjected to an aging process at approximately 250°C for one week in air in order to ensure stability, since salicylic acid is reported to decompose below 250°C.
Q: Lack of stabilization is a source of instability in sensor characteristics.
Answer: The lack of stability is indeed the biggest source of unstable sensor characteristics. In order to achieve stability, structural stability is first achieved, which is one of the reasons why we perform moderate aging processes. Nevertheless, there is still some distance from the ideal stability. But these deviations will not affect the conclusion of this work.
Q: In addition, without annealing, there is a high probability of the presence of hydroxides with physicochemical properties different from those of oxides.
Answer: Yes, there is a high probability that hydroxide will be present in the sensor material without annealing. These will affect the performance of the sensor. This is one of the reasons why we perform moderate aging processes.
How was the temperature controlled with such accuracy?
Answer: The measurement temperature is controlled by adjusting the voltage of the heater. The accuracy of temperature control is directly limited by the accuracy of the voltage source. The purpose of this test is a basic research in principle, the trend is the most important. The accuracy of temperature control is about 2 degrees.
What was the source of acetone vapor? How was the gas mixture formed?
Answer: There are two preparation methods for different acetone concentrations: (1) Preparation of high concentration acetone (10 ppm to 1000 ppm): acetone vapor directly comes from acetone solution. Accord to the calculation, the micro syringe injects accuracy concentrations of acetone liquid into the test chamber (250 ml volume) through a rubber stopper, and forms acetone vapor with designed concentration after heating. (2) Preparation of low concentration (50 ppb-2 ppm) acetone: in order to ensure the accuracy of gas concentration, two mass flow meters (MF) are used, one is acetone and the other is high air. An adjustment for the ratio of two gases flow ratio to form the getting designed acetone gas concentration.
It is known that at low temperatures humidity has a significant effect on sensors characteristics. However, this effect has not been evaluated and the constancy of humidity during the measurement process is not ensured. This is especially important for sensors designed to analyze the composition of exhaled air (the breathing gas).
Answer: The reviewer made a very good point, thanks. Humidity is always a very important factor affecting the performance of metal oxide-based gas sensors. Numerous theoretical and experimental data have been established. Therefore, measurement and borrowing those theory and data should be considered in subsequent optimization experiments. Actually, the performance of the sensors was measured at the humidity of 35%. The related the information was written at line 182 of revised manuscript.
How was the increase in sensitivity of XPS measurements achieved when measuring the XPS spectra of Pt? Why the Pt concentration was not estimated by XPS?
Answer: We have added the deconvolution of Pt peaks in Figure 5c. And the discussion about it has been added in the modified manuscript as follows (line 262-267): XPS shows a Pt concentration of 0.21 at%. We added a simple description at line 274.
Figure 5c. XPS spectra of Pt 4f electron binding energies of Pt-doped 3DPS.
The Pt 4f (Figure 5c) signals display two pairs of doublets, indicating the presence of two oxidation states. The most intense doublet with binding energy of 73.3 eV (Pt 4f7/2) and 76.6 eV (Pt 4f5/2) is attributed to Pt2+ as in PtO. The second doublet found at 74.1 and 77.6 eV appears to be Pt4+, possibly as PtO2 [28-30]. In contrast, metallic Pt peaks (70-71 eV) were not detected in the crystal structure of Pt-doped 3DPS in Figure 5c[31].
[28]Hernández-Fernández, P.; Nuño, R.; Fatás, E.; Fierro, J. L. G.; et al. MWCNT-supported PtRu catalysts for the electrooxidation of methanol: effect of the functionalized support. Int. J. Hydrogen Energy 2011, 36, 8267-8278.
[29]Yasuda, K.; Nobu, M.; Masui, T.; Imanaka, N.; et al. Complete oxidation of acetaldehyde on Pt/CeO2–ZrO2–Bi2O3 catalysts. Mater. Res. Bull. 2010, 45, 1278-1282.
[30] Zheng, Y.; Qiao, J.; Yuan, J.; Shen, J.; et al. Controllable synthesis of PtPd nanocubes on graphene as advanced catalysts for ethanol oxidation. Int. J. Hydrogen Energy 2018, 43, 4902-4911.
[31]Romanovskaya, V.; Ivanovskaya, M.; Bogdanov, P. A study of sensing properties of Pt-and Au-loaded In2O3 ceramics. Sens Actuators, B 1999, 56, 31-36.
Why describe in detail results what is not used in the analysis of the main results?
Answer: Thanks for the reviewer’s insight. we restated the theoretical analysis link with the relevant experimental data in sensing mechanism discussion section.
What is the thickness of the sensitive layer?
Answer: The thicknesses of sensitive layer film is estimated to be about 5 um.
What are the temperature conditions for manufacturing sensors?
Answer: At room temperature, the sensitive material is coated on the surface of the electrode. And then dry the sensor in an oven at 60 °C. The sensor is loaded on the aging table and aged at 250 °C for about a week.
Many questions to discuss the results.
When you look at the data provided by the authors for a sensor response with an accuracy of tenths such as 70.9, 143.1 or 505.7, then you immediately become convinced that the authors measured once on three different samples and therefore they have no data regarding the reproducibility of the results. If we repeat the process of synthesis and manufacturing of sensors, will the same results be obtained?
Answer: The reviewer made an effective point, thanks. We usually perform repeatability tests on all samples. As shown in the figure 8f at the manuscript, four cycles of measurement, at a temperature of 153 °C and exposed to 100 ppm acetones, showed excellent repeatability.
Figure 2. Repeatability of the sensor based on Pt-doped 3DPS after five-cycles of exposure to 100 ppm acetones at 153°C.
How can we talk about the maximum sensitivity at 153°C, if the measurements were carried out in increments of 25-50°C? Moreover, apparently, the temperature is not uniform over the area of the sensor chip.
Answer: It is an interesting question, many thanks! 153°C is not a very accuracy temperature. As we answered your question above, the measurement temperature is controlled by adjusting the voltage of the heater. The accuracy of temperature control is directly limited by the accuracy of the voltage source. The purpose of this test is a basic research in principle, the trend is the most important. The accuracy of temperature control is about 2 degrees. Therefore, we would like to guess that the optimal temperature should be 153+/- 2°C.
The reason for such a slow sensory response is not clear? Is this related to the properties of the material or to the peculiarity of the measuring system? Otherwise, there is no need to talk about the kinetics of the sensory response. Did you control the time of the change of atmosphere in the measuring chamber?
Answer: The reviewer made an effective point, thanks. We should say that any comparison among response time is non-meaningful if the characterizing chamber is not at same size. Response time fully depends on the nature of the material and the test system. In a static test system, the gas needs sufficient time to reach the set concentration. At the same time, it expected to take a long time for the porous material to adsorb up to saturation. That is why the response time takes long. Nevertheless, to compare response times among different materials are still meaningful, since the experimental procedures are same and response time may solely be resulting from the properties of the material. It is worth of mention that we may not worry about the effect of response time on the actual early diagnosis of the disease by measuring the breathing gas, because a heating or optically-assisted desorption system would be implemented to accelerate the desorption process.
Flower-like microstructure have very high gas permeability (porosity) and therefore it is not clear what kind of pores with a diameter of 10 nm exist in structures and how do they control the adsorption-desorption processes occurring in them?
Answer: We appreciated the reviewer’s comments. The purpose of generating micropores in the flower-like microstructure by annealing is to (1) increase the specific surface area and (2) provide a diffusion channel for the gas from the surface to the back. The purpose of these micropores is objective to enhance adsorption and increase the sensitivity of the gas sensor.
There is no answer to the question - what element of the structure determines the sensitivity? It is known that in such branched structures, many elements do not affect the sensor response, since they do not participate in current transport.
Answer: Thank you for the reviewer's question. Answering which element of the structure determines sensitivity is a difficult question. There are many factors that affect the sensitivity of metal oxide-based semiconductor sensors. There are some established theories. Basically, sensitivity involves the adsorption and reaction of molecules on the surface of sensitive materials. Any factor that can enhance the adsorption and reaction of molecules on the surface of the material can increase sensitivity. In this article, the purpose of synthesizing a flower-like structure is to increase the specific surface area. Pt-doping is to reduce the activation energy of adsorption and reaction to enhance adsorption and reaction. The micropores are created by the annealing process to increase specific surface area and diffusion channels. All of these efforts have improved material-to-gas pair sensitivity. Regards the elements of oxides, the adsorbed sites of preferential absorption play an important role in the sensitivity. Pt is most important site for molecular adsorption.
We agree with the statement that in such a branch structure, many elements may not affect the sensor response because they do not participate in current transmission. Although the contribution to sensitivity still comes from the surface of the materials, the key is that increasing the specific surface area can increase the opportunity for the molecules, after all, the adsorption and reaction only occurs on the surface of the material to contact.
What is the role of annealing in increasing sensitivity? Why did annealing increase the sensitivity despite the fact that the surface area appears to have decreased?
Answer: Thank you for the reviewer's question. The micropores are created by the annealing process to increase specific surface area and diffusion channels. All of these efforts have improved material-to-gas pair sensitivity.
How do you explain such large clusters of platinum (10-20 nm) at such a low concentration (1%)?
Answer: Thanks for this insight question. According the size distribution diagram (Figure S1), the average size of platinum nanoparticles is estimated to be about 8-12 nm. For the Pt-doped 3DPS material, after annealing at 700°C, extensive agglomeration of the platinum nanoparticles may result in significant increase of particle size.
